# Self-Consuming Generative Models with Adversarially Curated Data

**Xiukun Wei** [1]   **Xueru Zhang** [1]

## Abstract

Recent advances in generative models have made it increasingly difficult to distinguish real data from model-generated synthetic data. Using synthetic data for successive training of future model generations creates "self-consuming loops," which may lead to model collapse or training instability. Furthermore, synthetic data is often subject to human feedback and curated by users based on their preferences. Ferbach et al. (2024) recently showed that when data is curated according to user preferences, the self-consuming retraining loop drives the model to converge toward a distribution that optimizes those preferences. However, in practice, data curation is often noisy or adversarially manipulated. For example, competing platforms may recruit malicious users to adversarially curate data and disrupt rival models. In this paper, we study how generative models evolve under self-consuming retraining loops with noisy and adversarially curated data. We theoretically analyze the impact of such noisy data curation on generative models and identify conditions for the robustness of the retraining process. Building on this analysis, we design attack algorithms for competitive adversarial scenarios, where a platform with a limited budget employs malicious users to misalign a rival's model from actual user preferences. Experiments on both synthetic and real-world datasets demonstrate the effectiveness of the proposed algorithms.

## 1. Introduction

The latest generative models can produce highly realistic texts (OpenAI, 2024), images (Diffusion, 2025), audio (AI, 2025), and videos (ML, 2025). As synthetic data proliferates on the internet, it is inevitably used for training future generations of models, creating a "self-consuming training loop." A line of research has focused on examining the impact of this self-consuming training loop on generative models' outputs, both theoretically (Bertrand et al., 2024; Taori & Hashimoto, 2023) and empirically (Gerstgrasser et al., 2024; Alemohammad et al., 2024a; Shumailov et al., 2024). These studies demonstrate that such a self-consuming training loop may lead to model collapse (Gerstgrasser et al., 2024; Alemohammad et al., 2024a; Shumailov et al., 2024), training instability (Bertrand et al., 2024), and possibly bias amplification (Taori & Hashimoto, 2023; Wyllie et al., 2024; Xie & Zhang, 2024). Several solutions have also been proposed to mitigate these issues, such as integration of real data (Bertrand et al., 2024), leveraging cumulative datasets (Gerstgrasser et al., 2024), and employing Self-Improving Diffusion Models with Synthetic Data (SIMS) (Alemohammad et al., 2024b).

In contrast to these works, a recent study (Ferbach et al., 2024) explores a more practical scenario in which synthetic data is curated by human users. To improve safety, user trust, and the relevance and quality of generated outputs, modern generative models are increasingly trained with human participation and feedback. For example, platforms such as JourneyDB (Pan et al., 2023) and Pika Labs (Labs, 2025) provide multiple variations of outputs for users to choose from, with only selected outputs being upscaled and used to train next-generation models. As shown in Ferbach et al. (2024), when synthetic data is curated based on a reward model representing user preferences, the generative models trained iteratively on this curated data tend to converge to an output distribution that maximizes the expected reward.

However, in practice, data curated from users are likely to be noisy, biased, or even maliciously manipulated. Consider a scenario where multiple platforms compete for the same target user population with similar preference distributions (e.g., ChatGPT (OpenAI, 2025) and Claude (Anthropic, 2025), Stable Diffusion (Diffusion, 2025) and MidJourney (MidJourney, 2025)). To compete for market share, platforms may leverage their collected datasets, which contain rich information about actual user preferences, to design attack algorithms targeting their competitors. For example, as shown in Fig. 1, a platform may employ malicious users with limited budgets to deliberately select outputs on a ri-

---

[1]Department of Computer Science and Engineering, The Ohio State University, Columbus, Ohio, USA. Correspondence to: Xueru Zhang <zhang.12807@osu.edu>.

*Proceedings of the 42nd International Conference on Machine Learning*, Vancouver, Canada. PMLR 267, 2025. Copyright 2025 by the author(s).

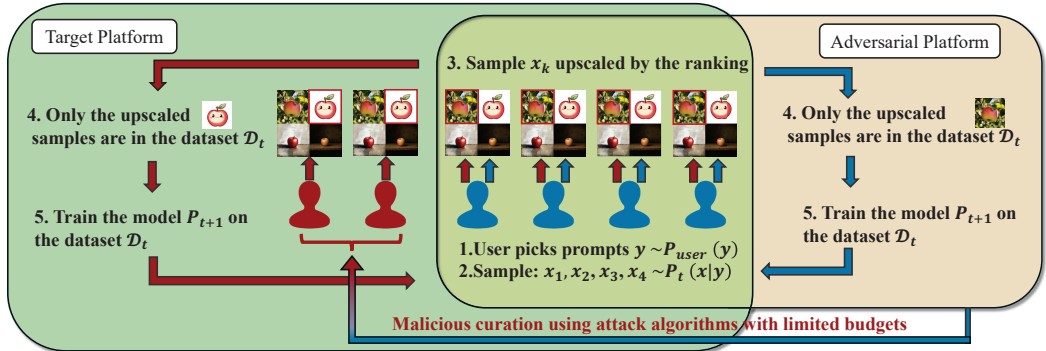

*Figure 1.* An example of adversarial data curation in a competitive setting: the adversarial platform and the target platform serve the same population of users. The adversarial platform can access the data generated by the target platform and obtain real user preferences through its own preference collection mechanisms. Using the attack algorithm, the adversarial platform employs malicious users to adversarially curate data on the target platform, preventing its model from aligning with genuine user preferences.

val's platform that significantly deviate from genuine user preferences, creating a dataset on the competing platform that no longer reflects true user preferences. Over time, this adversarially curated data degrades the competitor's ability to train models that align with user preferences, ultimately reducing their capacity to produce content that attracts users.

This paper takes a first step in examining the impact of adversarially curated data on the iterative retraining of generative models. We provide a theoretical analysis to understand how the output distribution of generative models evolves under adversarially curated data and assess the robustness of the self-consuming training loop to such manipulations. Our results demonstrate that under specific conditions, the self-consuming generative model trained from such adversarially curated data remains robust that it still converges to output distribution that optimizes user preferences. However, we also identify conditions (on the fraction of adversarial data and its associated reward function) under which this robustness guarantee fails to hold.

Building on this theoretical understanding, we design attack algorithms for adversarial data curation aimed at disrupting the alignment of self-consuming generative models with user preferences. Specifically, we consider a competitive scenario in which platforms compete for market share by employing malicious users to curate adversarial data on a rival's platform. Given a dataset reflecting genuine user preferences (collected from the platform's own users), our attack algorithms strategically flip a limited number of preference labels. The modified dataset then guides malicious users in curating adversarial data on the rival platform. Over time, the generative models on the targeted platform, when iteratively trained on such adversarially curated data, fail to align with genuine user preferences, reducing their ability to remain appealing to users. To the best of our knowledge, this is the first attack algorithm for deviating self-consuming generative models from user preference. The most related

work is Wu et al. (2025), which investigates the vulnerability of reward model learning to preference poisoning. However, unlike our work that focuses on a self-consuming training loop, where the attacker aims to gradually misalign the target model with human preferences, Wu et al. (2025) considers a static setting where the attacker aims to flip preference labels to promote or demote specific target outcomes under the learned reward model. In Appendix A, we discuss more related works.

Our contributions are summarized as follows:

- In Section 3, we theoretically analyze the long-term performance of self-consuming generative models under adversarial data curation, considering both pure synthetic data and mixtures of synthetic and real data.

- In Lemma 3.3, we prove that the convergence of the generative model toward maximizing user-expected rewards is governed by the covariance of adversarially curated synthetic data. This result establishes conditions where convergence remains robust and conditions where adversarial data curation hinders model alignment with human preferences.

- In Section 4, we model a competitive situation where an adversarial platform aims to disrupt a competitor's model alignment through adversarial data curation, and propose gradient-based and heuristic attack algorithms.

- In Section 5, we validate the theorems and proposed algorithms through experiments on synthetic and real datasets.

## 2. Problem Formulation

Consider a platform that iteratively trains generative models from data curated by users. Denote $p_{\text{data}} \in \mathcal{P}(\mathbb{R}^d)$ as the real data distribution and for $t \in \mathbb{N}$, let $p_t \in \mathcal{P}(\mathbb{R}^d)$ be the data distribution of the generative model at $t$-th round of iterative retraining loop. Throughout the paper, we use

lowercase letters $p$ to denote densities and uppercase letters $\mathbb{P}$ to indicate the associated probabilities.

**Adversarially curated data.** At every round $t \in \mathbb{N}$, the platform presents synthetic data $x_1, \cdots, x_K$ randomly sampled from current model $p_t$ to users, and users select their preferred outputs that will be upscaled in the dataset for training next-generation of models. Following Ferbach et al. (2024), we adopt the generalized Bradley-Terry model (Bradley & Terry, 1952) to model user's choice. Specifically, let $r(x)$ be an underlying reward function that captures user preference of one data $x_i$ over another $x_j$, then the probability that a data $\hat{x} \in \{x_1, \cdots, x_K\}$ is curated by the user is as follows:

$$\mathbb{P}\left(\hat{x} = x_k \mid x_1, \ldots, x_K\right) = \frac{e^{r(x_k)}}{\sum_{j=1}^{K} e^{r(x_j)}}, \quad k \in [K] \quad (1)$$

Because curated data in practice may be noisy and adversarially manipulated, we use a mixture model to account for such adversarial behavior. Specifically, assume another competing platform employs malicious users to curate noisy data deviating from actual user preference. Let $\phi_t \in (0, 1)$ be the fraction of malicious users at round $t$ and $\widetilde{r}_t(x)$ the underlying reward function. Then the probability $\hat{x}$ is curated by a mixture of malicious and benign users is:

$$\mathbb{P}(\hat{x} = x_k \mid x_1, \ldots, x_K) = (1 - \phi_t) \frac{e^{r(x_k)}}{\sum_{j=1}^{K} e^{r(x_j)}} \quad (2)$$
$$+ \phi_t \frac{e^{\widetilde{r}_t(x_k)}}{\sum_{j=1}^{K} e^{\widetilde{r}_t(x_j)}}, k \in [K]$$

We use $\hat{x} \sim \mathcal{BT}(x_1, \ldots, x_K; \phi_t)$ to denote $\hat{x}$ is sampled from $x_1, \ldots, x_K$ according to probability (2).

**Self-consuming training loop.** Given adversarially curated data, the platform updates its model at round $t+1$ according to the following, either solely on the distribution of curated synthetic data ($\lambda \to \infty$) or on a mixture of synthetic and real data ($\lambda < \infty$):

$$p_{t+1} = \arg\max_{p \in \mathcal{P}} \frac{1}{1+\lambda} \mathbb{E}_{x \sim p_{\text{data}}} \left[\log\left(p\left(x\right)\right)\right]$$
$$+ \frac{\lambda}{1+\lambda} \mathbb{E}_{\substack{x_1, \cdots, x_K \sim p_t \\ \hat{x} \sim \mathcal{BT}(x_1, \ldots, x_K; \phi_t)}} \left[\log\left(p\left(\hat{x}\right)\right)\right] \quad (3)$$

where $\lambda \in [0, \infty)$ controls the fraction of real data and $\mathcal{P}$ is the set of achievable distributions with the platform's model.

**Objectives.** A recent study (Ferbach et al., 2024) focused on a special case without malicious users ($\phi_t = 0$). They showed that if data are curated based on the benign users' reward function $r(x)$ and the generative model is updated solely using curated synthetic data, the self-consuming training loop can result in $p_t$ converging to a distribution that optimizes user preferences, i.e., as $t \to \infty$,

$\mathbb{E}_{x \sim p_t}[e^{r(x)}]$ converges to the maximum reward, and its variance $\text{Var}_{x \sim p_t}[e^{r(x)}]$ vanishes. However, in the presence of malicious users, it remains unclear how the model evolves and whether $p_t$ can align with actual user preferences in the long run. This paper explores this problem and we first examine the long-term impact of adversarially curated data on self-consuming models (Section 3) and then design attack algorithms that platforms can use to disrupt competitors' model training processes and misalign their models from user preferences (Section 4).

## 3. Impact of adversarially curated data

Next, we examine the evolution of self-consuming generative models with adversarially curated data. We begin with the case of purely synthetic data ($\lambda \to \infty$) and then generalize to a mixture of synthetic and real data ($\lambda < \infty$). All proofs can be found in Appendix B.

**Iterative retraining only on curated synthetic data.** Without real data, the iterative retraining process reduces to:

$$p_{t+1} = \arg\max_{p \in \mathcal{P}} \mathbb{E}_{\substack{x_1, \cdots, x_K \sim p_t \\ \hat{x} \sim \mathcal{BT}(x_1, \ldots, x_K; \phi_t)}} \left[\log\left(p\left(\hat{x}\right)\right)\right] \quad (4)$$

**Lemma 3.1.** *Consider the asymptotic case where the number of samples users select from satisfies $K \to \infty$. Suppose $\mathbb{E}_{x \sim p_t}[e^{r(x)}] < \infty$ and $p_{t+1}$ follows Eq. (4). Then, we have*

$$p_{t+1}(x) \to p_t(x) \left[(1 - \phi_t) \frac{e^{r(x)}}{\mathbb{E}_{z \sim p_t}\left[e^{r(z)}\right]} + \phi_t \frac{e^{\widetilde{r}_t(x)}}{\mathbb{E}_{z \sim p_t}\left[e^{\widetilde{r}_t(z)}\right]}\right]$$

Lemma 3.1 characterizes the relation between $p_{t+1}$ and $p_t$ in a self-consuming loop with adversarially curated data. Next, we analyze the evolution of $\mathbb{E}_{p_t}[e^{r(x)}]$, which quantifies the expected reward users experience from interacting with the generative model. We first introduce some technical assumptions similar to Ferbach et al. (2024).

**Assumption 3.2.** *There exist finite constants $r_{t,\min}, r_{t,\max}, \widetilde{r}_{t,\min}, \widetilde{r}_{t,\max} \in \mathbb{R}$, such that: $p_t$-almost surely, $\forall x \sim p_{\text{data}}$, $r_{t,\min} = \inf_x r(x)$, $r_{t,\max} = \sup_x r(x)$, $\widetilde{r}_{t,\min} = \inf_x \widetilde{r}_t(x)$, $\widetilde{r}_{t,\max} = \sup_x \widetilde{r}_t(x)$.*

In most realistic scenarios, this assumption holds because user preferences typically have finite support or are bounded in a probabilistic sense. Under this assumption, Lemma 3.3 below examines the impact of adversarially curated data on $\mathbb{E}_{p_{t+1}}[e^{r(x)}]$ and presents its upper and lower bounds.

**Lemma 3.3.** *Let $p_{t+1}$ be the distribution induced from a discrete choice model in Eq. (4). Suppose Assumption 3.2 holds, then the following holds:*

$$\mathbb{E}_{p_{t+1}}\left[e^{r(x)}\right] \geq \mathbb{E}_{p_t}\left[e^{r(x)}\right] + \frac{\widetilde{\text{Var}}}{e^{r_{t,\max}}} + \frac{\widetilde{\text{Cov}}}{e^{\widetilde{r}_{t,\min}}}$$

$$\mathbb{E}_{p_{t+1}}\left[e^{r(x)}\right] \leq \mathbb{E}_{p_t}\left[e^{r(x)}\right] + \frac{\widetilde{\text{Var}}}{e^{r_{t,\min}}} + \frac{\widetilde{\text{Cov}}}{e^{\widetilde{r}_{t,\max}}}$$

*where*

$$\widetilde{\mathbf{Var}} := (1 - \phi_t) \frac{(K-1)}{K} \operatorname{Var}_{p_t}\left[e^{r(x)}\right]$$

$$\widetilde{\mathbf{Cov}} := \phi_t \frac{(K-1)}{K} \operatorname{Cov}_{p_t}\left[e^{r(x)}, e^{\widetilde{r}_t(x)}\right]$$

According to Lemma 3.3, when $e^{r(x)}$ and $e^{\widetilde{r}_t(x)}$ are positively correlated, i.e., $\operatorname{Cov}_{p_t} \geq 0$, the expected reward increases, $\mathbb{E}_{p_{t+1}}\left[e^{r(x)}\right] \geq \mathbb{E}_{p_t}\left[e^{r(x)}\right]$, allowing the generative model to align with user preferences despite adversarially curated data. In this case, the expected reward converges to the maximum, highlighting the model's inherent **robustness** against noise and adversarial attacks. However, when $e^{r(x)}$ and $e^{\widetilde{r}_t(x)}$ are negatively correlated, i.e., $\operatorname{Cov}_{p_t} < 0$, the convergence is no longer guaranteed. Instead, the expected reward may oscillate and deviate from the maximum value. This shows the model's potential **vulnerability** in scenarios where adversarially curated data introduces negative correlations with the user reward function. The rationale for the upper bound is discussed in Appendix B.3.

**Iterative retraining on mixed real and synthetic data.** Prior works such as Ferbach et al. (2024); Bertrand et al. (2024); Gerstgrasser et al. (2024); Alemohammad et al. (2024a) explored the role of real data in self-consuming generative models. Without data curation, Bertrand et al. (2024); Gerstgrasser et al. (2024); Alemohammad et al. (2024a) showed that retraining models with a mix of real and synthetic data help stabilize the algorithm and prevent $p_t$ from deviating too much from $p_{\text{data}}$. When synthetic data is curated by users based on their preferences, $p_{\text{data}}$ is no longer a fixed point of the retraining loop, as different reward values can occur with positive probability; however, $\mathbb{E}_{p_t}[e^{r(x)}]$ still increases compared to $\mathbb{E}_{p_{\text{data}}}[e^{r(x)}]$ (Ferbach et al., 2024). Next, we study whether incorporating real data can help defend against adversarially curated data during iterative model retraining.

**Lemma 3.4.** *Let $p_{t+1}$ be defined as in Eq. (3), with $p_0 = p_{data}$ and $\phi_t = \phi_\star, \forall t$. The following holds for all $t$:*

$$\mathbb{E}_{p_{t+1}}\left[e^{r(x)}\right] \geq \mathbb{E}_{p_{data}}\left[e^{r(x)}\right] \tag{5}$$
$$+ \phi_\star(1+\lambda)\left(1 - \left(\frac{\lambda}{1+\lambda}\right)^t\right)\operatorname{Cov}_{\min}$$

*where* $\operatorname{Cov}_{\min} = \min_{i \in [t]} \operatorname{Cov}_{p_i}\left[e^{r(x)}, e^{\widetilde{r}_i(x)}\right]$. *As* $t \to \infty$,

$$\mathbb{E}_{p_{t+1}}\left[e^{r(x)}\right] \geq \mathbb{E}_{p_{data}}\left[e^{r(x)}\right] + \phi_\star(\lambda+1)\operatorname{Cov}_{\min} \tag{6}$$

Lemma 3.4 implies that when $\operatorname{Cov}_{\min} > 0$, i.e., the adversarial reward values are positively correlated with the true rewards, model can align with genuine user preferences

and $\mathbb{E}_{p_t}[e^{r(x)}] > \mathbb{E}_{p_{\text{data}}}[e^{r(x)}]$ still holds. However, when $\operatorname{Cov}_{\min} < 0$, this is no longer the case, suggesting that simply adding real data is not sufficient to defend against adversarial curation.

## 4. Attack algorithms

As shown in Lemma 3.3, adversarially curated data may result in $\mathbb{E}_{x \sim p_t}[e^{r(x)}]$ deviating from maximum reward in the long run. Next, we consider a competitive scenario illustrated in Fig. 1, where an *adversarial platform* employs malicious users to curate data on a *target platform* to compete for market share. Our goal is to design attack algorithms for the adversarial platform that guide malicious users to act in the most effective way to disrupt the target platform's model.

**Learning reward model from pairwise comparisons.** Unlike Ferbach et al. (2024) that assumes user reward model $r(x)$ is known, we consider a more realistic setting that the adversarial platform does not have access to $r(x)$ but must learn it from a dataset indicating user preferences. Here, we consider learning from pairwise comparisons (Wu et al., 2025; Liu et al., 2024; Zhou et al., 2025) as detailed below.

Let $\mathcal{D} = \{(x_i, z_i, o_i)\}_{i=1}^n$ be the dataset adversarial platform collects from benign users, where $x_i, z_i \in \mathbb{R}^d$ are $i$-th pair of data samples acquired from the target platform and $o_i \in \{0, 0.5, 1\}$ indicates the user's preference among them[1]. Specifically, $o_i = 0$ if $x_i$ is preferred to $z_i$ ($x_i \succ z_i$), $o_i = 1$ if $z_i$ is preferred to $x_i$ ($x_i \prec z_i$), and $o_i = 0.5$ if $y_i$ and $z_i$ are equally preferred. In this paper, we assume adversarial and target platforms face users with identically distributed preferences, so that the adversarial platform can learn $r(x)$ from $\mathcal{D}$.

Let $R_\theta$ be a parametric reward model of $r(x)$ learned from preference data $\mathcal{D}$, where $\theta \in \Theta$ is the parameter. A typical method is Maximum Likelihood Estimation (MLE) which minimizes the following loss function:

$$\mathcal{L}(\mathcal{D}; \theta) = -\sum_i \left[(1 - o_i)\log \Pr\{x_i \succ z_i \mid R_\theta\}\right.$$
$$\left. + o_i \log \Pr\{z_i \succ x_i \mid R_\theta\}\right] \tag{7}$$

where preference label $o_i$ is generated according to

$$o_i \sim \Pr\{z_i \succ x_i \mid r\} = \frac{e^{r(z_i)}}{e^{r(x_i)} + e^{r(z_i)}} \tag{8}$$

**Objective and constraint of the adversarial platform.** To compete against the target platform, the adversarial platform

---

[1]To obtain this dataset, the adversarial platform can first acquire $K$ data samples $\{x_1, \cdots, x_K\}$ from target platform and then present them to its users to select. Suppose $K = 3$ and the user selects $x_2$, the data pair $\{(x_1, x_2, 1), (x_2, x_3, 0)\}$ is added to $\mathcal{D}$.

employs malicious users to adversarially curate data on the target platform, aiming to cause the generative models iteratively trained on this curated data to deviate from user preferences. The behavior of malicious users can be formalized as "flipping the preference label $o_i$ of data pairs." Specifically, given $\mathcal{D} = \{(x_i, z_i, o_i)\}_{i=1}^n$, malicious users can flip preference $o_i$ to $o_i + \delta_i$, resulting in an adversarial dataset $\widetilde{\mathcal{D}}(\boldsymbol{\delta}) = \{(x_i, z_i, o_i + \delta_i)\}_{i=1}^n$, where $\delta_i = \{-1, 0, 1\}$ presents label perturbation and $\boldsymbol{\delta} := \{\delta_i\}_{i=1}^n$. In practice, there is a possibility that malicious users fail to curate data adversarially on the target platform, or the curated data is not selected for training the future model. To account for this, we impose a constraint on the total number of label flips $\sum_{i=1}^n |\delta_i| \leq \kappa \cdot n$, where $\kappa \in (0, 1)$ represents the success rate of perturbing the data on the target platform.

The goal of the adversarial platform is to find $\boldsymbol{\delta}$ such that the resulting perturbed dataset $\widetilde{\mathcal{D}}(\boldsymbol{\delta})$, when mixed with benign users' preference data (via malicious users' adversarial data curation), disrupts the target platform's alignment with user preferences. Formally, let $p_{t+1}$ be the self-consuming model of the target platform trained from such adversarially curated data, the goal is to reduce the expected reward $\mathbb{E}_{x \sim p_t}[e^{r(x)}]$ such that it can deviate from the maximum reward in the long run, i.e.,

$$\mathbb{E}_{p_{t+1}}\left[e^{R_\theta(x)}\right] < \mathbb{E}_{p_t}\left[e^{R_\theta(x)}\right] \tag{9}$$

where $R_\theta$ is parametric reward model learned from benign users. Based on Lemma 3.3, we formulate the following optimization for adversarial platform:

$$\min_{\boldsymbol{\delta}} \mathcal{J}(\boldsymbol{\delta}) := \mathrm{Cov}_{p_t}\left[e^{R_\theta(x)}, e^{\widetilde{R}_{\widetilde{\theta}}(x)}\right] + \alpha \operatorname{dist}\left(R_\theta, \widetilde{R}_{\widetilde{\theta}}\right)$$
$$\text{s.t.} \quad \widetilde{\theta} \in \arg\min_{\theta'} \mathcal{L}\left(\widetilde{D}(\boldsymbol{\delta}), \theta'\right) \tag{10}$$

where $\widetilde{R}_{\widetilde{\theta}}$ is parametric reward model learned from perturbed preference data $\widetilde{D}(\boldsymbol{\delta})$, which may belong to a different function family than $R_\theta$. To achieve the objective in Eq. (9), we aim to make $\mathrm{Cov}_{p_t}[e^{R_\theta(x)}, e^{\widetilde{R}_{\widetilde{\theta}}(x)}]$ as negative as possible. Meanwhile, to prevent the adversarial behavior from being easily detected as anomalous, we impose a penalty on the difference between $\widetilde{R}_{\widetilde{\theta}}$ and $R_\theta$, quantified by $\operatorname{dist}(R_\theta, \widetilde{R}_{\widetilde{\theta}})$ and can be defined as $\mathbb{E}_{p_t}[d(R_\theta(x), \widetilde{R}_{\widetilde{\theta}}(x))]$ for some distance metric $d$ (e.g., $\ell_p$ norm).

**Dynamic attack during iterative training.** Since the target platform dynamically updates its model over time, the adversarial platform must repeatedly solve optimization (10) as $p_t$ evolves. In practice, the adversarial platform can periodically interact with the target platform to acquire data pairs $x_i^{(t)}, z_i^{(t)} \sim p_t$ and collect user preference $o_i^{(t)}$ from its own customers. The dataset $\mathcal{D}^{(t)} = \{(x_i^{(t)}, z_i^{(t)}, o_i^{(t)})\}_{i=1}^n$ can be first used to fine-tune benign reward model $R_\theta$ and

then solve for $\boldsymbol{\delta}^{(t)}$ in optimization (10). The resulting $\boldsymbol{\delta}^{(t)}$ can then guide the malicious users in curating data on the target platform at $t$. Such adversarially curated data is subsequently used by the target platform to retrain its model $p_{t+1}$. Over time, this iterative interaction may cause the target platform to deviate significantly from user preferences.

**Challenges to solve optimization (10).** The solution to (10) is difficult to find because it is a *bi-level* optimization problem. Moreover, the variables to be optimized are a subset of preference comparison labels, which involves solving a *combinatorial* optimization problem over a discrete space. Next, we tackle these challenges and introduce two methods for finding approximated solutions to optimization (10).

### 4.1. Gradient-based methods

To tackle discrete decision space, we relax the action space $\delta_i \in \{-1, 0, 1\}$ to interval $\delta_i \in [-1, 1]$. If reward models are differentiable with respect to $\widetilde{\theta}$, then we can compute the gradient of the objective function in (10) as follows:

$$\nabla_{\boldsymbol{\delta}} \mathcal{J}(\boldsymbol{\delta}) = \nabla_{\boldsymbol{\delta}} \left(\mathrm{Cov}_{p_t}\left[e^{R_\theta(x)}, e^{\widetilde{R}_{\widetilde{\theta}}(x)}\right] + \alpha \operatorname{dist}\left(R_\theta, \widetilde{R}_{\widetilde{\theta}}\right)\right)$$

$$= \nabla_{\widetilde{\theta}} \left(\mathrm{Cov}_{p_t}\left[e^{R_\theta(x)}, e^{\widetilde{R}_{\widetilde{\theta}}(x)}\right] + \alpha \operatorname{dist}\left(R_\theta, \widetilde{R}_{\widetilde{\theta}}\right)\right) \frac{d\widetilde{\theta}}{d\boldsymbol{\delta}}$$

Recall that $\widetilde{R}_{\widetilde{\theta}}$ is the reward model trained from perturbed dataset $\widetilde{\mathcal{D}}(\boldsymbol{\delta})$, parameter $\widetilde{\theta}$ is a function of $\boldsymbol{\delta}$. Similar to Wu et al. (2025), we can leverage the implicit function theorem (Mei & Zhu, 2015; Koh & Liang, 2017) to compute the implicit derivative $\frac{d\widetilde{\theta}}{d\boldsymbol{\delta}}$:

$$\frac{d\widetilde{\theta}}{d\boldsymbol{\delta}} = -\left[H_{\widetilde{\theta}}\mathcal{L}\right]^{-1}\left[\frac{d\nabla_{\widetilde{\theta}}\mathcal{L}}{d\boldsymbol{\delta}}\right] \tag{11}$$

where $H_{\widetilde{\theta}}\mathcal{L}$ is the Hessian of $\mathcal{L}$ with respect to $\widetilde{\theta}$. Given the gradient, we can then apply projected gradient descent to find optimal $\boldsymbol{\delta}^*$. Since original action space is $\delta_i \in \{-1, 0, 1\}$ and the total number of flips is constrained by $\sum_{i=1}^n |\delta_i| = \kappa \cdot n$, we select the the top $\kappa \cdot n$ among $\boldsymbol{\delta}^*$ based on magnitude $|\delta_i^*|$ and only flip $o_i$ associated to them. The complete procedure is shown in Algorithm 1.

The efficiency of this algorithm is primarily influenced by two key factors: batch size and model structure. Batch size plays a critical role in the accuracy of covariance estimation, which is essential for aligning the computed covariance with the true data distribution. While larger batch sizes generally yield more precise covariance estimates and reduce the discrepancy between theoretical and practical outcomes, they also incur greater memory requirements and computational overhead. Meanwhile, the structure of the initialized model directly influences the size of the implicit Hessian matrix, which emerges during gradient computations involving second-order derivatives. These calculations are

---

**Algorithm 1** Gradient-based attack

  **Input:** Benign preference data $\mathcal{D}$, parameter $\kappa$
  Train the reward model $R_\theta$ of benign users on $\mathcal{D}$;
  Randomly initialize $\boldsymbol{\delta}$, ensuring $(o_i + \delta_i) \in [-1, 1]$;
  **for** $m = 1$ **to** $M$ **do**
    Create a perturbed dataset $\widetilde{\mathcal{D}}(\boldsymbol{\delta})$;
    Train a new reward model $\widetilde{R}_{\widetilde{\theta}}$ on the dataset $\widetilde{\mathcal{D}}(\boldsymbol{\delta})$;
    Compute the gradient $\nabla_{\boldsymbol{\delta}} \mathcal{J}(\boldsymbol{\delta})$;
    $\boldsymbol{\delta} \leftarrow \boldsymbol{\delta} - \eta \nabla_{\boldsymbol{\delta}} \mathcal{J}(\boldsymbol{\delta})$;
    Clip $\boldsymbol{\delta}$ such that $(o_i + \delta_i) \in [-1, 1], \forall i \in [n]$;
  **end for**
  Select the top $\kappa \cdot n$ indices based on $|\delta_i|$;
  Flip the preference label of the corresponding data pair;
  **Output:** Label perturbations $\boldsymbol{\delta}$, reward model $\widetilde{R}_{\widetilde{\theta}}$

---

computationally expensive. Although approximate methods can be used to estimate the Hessian matrix and reduce the computational burden, models with complex architectures—such as deep neural networks with many layers and parameters—substantially increase both the computational time and memory requirements for these calculations.

### 4.2. Heuristic methods

The high computational costs associated with gradient-based calculations for complex models present a significant challenge. To mitigate this issue, we propose leveraging heuristic methods as an alternative. These methods eliminate the need for explicit gradient computation, providing a more computationally efficient approach, as detailed below.

**Reward-based heuristic method.** Instead of directly optimizing perturbations $\boldsymbol{\delta}$ to minimize $\mathrm{Cov}_{p_t}[e^{R_\theta(x)}, e^{\widetilde{R}_{\widetilde{\theta}}(x)}] + \alpha \, \mathrm{dist}(R_\theta, \widetilde{R}_{\widetilde{\theta}})$, we adopt a heuristic approach by flipping the preference label $o_i$ based on the rewards of data samples. Specifically, given preference data $\mathcal{D} = \{(x_i, z_i, o_i)\}_{i=1}^n$, we first learn the reward model $R_\theta$ from $\mathcal{D}$. The idea is to identify $\kappa \cdot n$ data pairs $(x_i, z_i)$ based on their rewards $R_\theta(x_i), R_\theta(z_i)$ such that flipping their preference label $o_i$ has the greatest impact on the underlying reward model.

We propose two methods for finding such data pairs: (i) finding $(x_i, z_i)$ based on dissimilarity between $R_\theta(x_i)$ and $R_\theta(z_i)$; (ii) finding $(x_i, z_i)$ based on maximum of $|R_\theta(x_i)|$ and $|R_\theta(z_i)|$. Specifically, define $f : \mathbb{R}^d \times \mathbb{R}^d \rightarrow \mathbb{R}^+$ as

$$f(x, z) := |R_\theta(x) - R_\theta(z)| \text{ or } \max\{|R_\theta(x)|, |R_\theta(z)|\}.$$

We select the $\kappa \cdot n$ data pairs $(x_i, z_i)$ with the highest $f(x_i, z_i)$ values to flip their preference label. Intuitively, a larger $|R_\theta(x_i) - R_\theta(z_i)|$ indicates greater differences in user preferences for the data pair $(x_i, z_i)$, making the preference flip more impactful. Similarly, a larger $\max\{|R_\theta(x_i)|, |R_\theta(z_i)|\}$ suggests that the pair includes a sample that is either highly favored or strongly disliked by

the user, making the preference flip most effective.

**Multi-objective heuristic method.** Since optimization problem (10) simultaneously considers two competing objectives, we also propose a heuristic method that finds the Pareto front (Ngatchou et al., 2005) for multi-objective optimization. Specifically, a solution is considered Pareto optimal if no other solution exists that can improve one objective without degrading at least one other objective; the set of all Pareto optimal solutions forms the Pareto front.

Our method begins by generating $\kappa \cdot n$ random perturbations $\boldsymbol{\delta}$ to flip data $\mathcal{D}$, resulting in $\widetilde{\mathcal{D}}(\boldsymbol{\delta})$. For each perturbation, we record the empirical performances of $\mathrm{Cov}_{p_t}[e^{R_\theta(x)}, e^{\widetilde{R}_{\widetilde{\theta}}(x)}]$ and $\mathrm{dist}(R_\theta, \widetilde{R}_{\widetilde{\theta}})$. Using a Pareto optimization algorithm, such as NSGA-II (Deb et al., 2002), we compute the Pareto front of non-dominated solutions, which represents the best trade-offs between objectives. Finally, we select the optimal solution from the Pareto front based on a specific prioritized objective, and apply the corresponding flip.

## 5. Experiments

In this section, we conduct experiments on both synthetic and real data to validate our theorems and proposed algorithms. We first evaluate the evolution of generative model $p_t$ and user reward $\mathbb{E}_{p_t}[r(x)]$ under self-consuming training loop (Section 5.1). Then, we demonstrate the effectiveness of proposed attack algorithms (Section 5.2).

**Datasets.** Similar to Ferbach et al. (2024), we conduct experiments on three datasets:

1. **Synthetic Gaussian**: A dataset following 8-mode Gaussian mixture model, the details are in Appendix C.2.

2. **CIFAR-10** (Krizhevsky, 2009): It contains 60,000 images from 10 classes {airplne := 0, automobile := 1, bird := 2, cat := 3, deer := 4, dog := 5, frog := 6, horse := 7, ship := 8, truck := 9}.

3. **CIFAR-100** (Krizhevsky, 2009): It contains 100 classes, each with 600 images. Class labels from 0 to 99 are assigned according to the alphabetical order of class names, i.e., {aquatic mammals-beaver := 0, $\cdots$, vehicles 2-tractor := 99}.

We present the results for CIFAR-10 and CIFAR-100 below, while the results for the Gaussian data are provided in the Appendix C.2.

**Reward functions and user preference labels.** Using Gaussian, CIFAR-10, and CIFAR-100 datasets, we can construct user preference dataset $\mathcal{D} = \{(x_i, z_i, o_i)\}_{i=1}^n$ using a reward function $r(x)$. Specifically, given a data pair $(x_i, z_i)$ sampled from Gaussian, CIFAR-10 or CIFAR-100 datasets, we generate the corresponding preference label $o_i \sim \Pr\{z_i \succ x_i \mid r\}$ based on Eq. (8). $r(x)$ for each

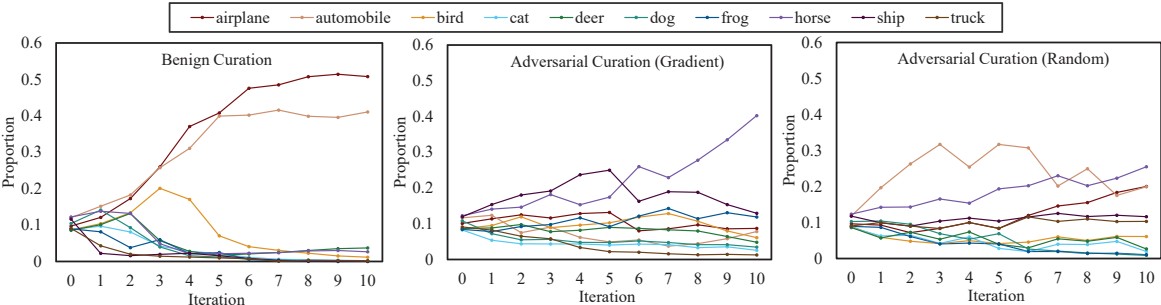

*Figure 2.* The proportion of each class generated by the self-consumption model retrained with different data curation methods on CIFAR-10: benign curation based on actual user preferences (left), adversarial curation using the proposed gradient-based attack algorithm (middle), and adversarial curation via a random attack (right). The results show that the proposed gradient-based attacks are the most effective in deviating the model from user preferences.

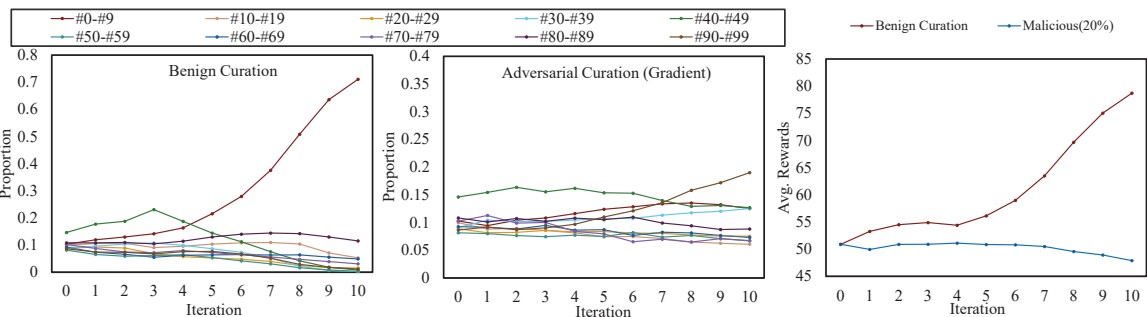

*Figure 3.* The proportion of each ten classes generated by the self-consumption model retrained with different data curation methods on CIFAR-100: benign curation based on actual user preferences (left), adversarial curation using the proposed gradient-based attack algorithm (middle). And empirical estimate of $\mathbb{E}_{p_t}[r(x)]$ from samples generated by the model over iterations (right).

dataset is defined as follows:

1. **Synthetic Gaussian:** $r(x) := -\gamma \max\{0, \|x - \mu_*\| - \tau\}$, where $\|x - \mu_*\|$ is the distance from one Gaussian center $\mu_*$, $\tau$ is the minimum clipped radius, and $\gamma$ controls the scaling of the reward (details are in Appendix C.2). Intuitively, this function captures user preferences by assigning higher rewards to samples farther from Gaussian center within a threshold.

2. **CIFAR-10:** First identify the label $I(x) \in \{0, \cdots, 9\}$ of image $x$ by a pretrained VGG11 (Simonyan & Zisserman, 2015) classifier with 92.79% test accuracy. Suppose users prefer classes with smaller indices and define $r(x) := 10 - I(x)$. It reflects user's preference ordering by assigning higher rewards to images classified closer to the most preferred class (class 0) in the hierarchy.

3. **CIFAR-100:** Similar to CIFAR-10, the label $I(x) \in \{0, \cdots, 99\}$ of image $x$ is first identified by a pretrained ResNet56 (He et al., 2016) classifier with 72.63% test accuracy. Suppose users prefer classes with smaller indices and define $r(x) := 100 - I(x)$.

**Generative model and reward model.** Following Ferbach et al. (2024), we iteratively retrain a denoising diffusion

probabilistic model (DDPM) (Ho et al., 2020), a generative framework known for its ability to model complex data distributions through a reversible diffusion process. In addition to the generative model, the target and adversarial platforms leverage user preference data $\mathcal{D} = \{(x_i, y_i, o_i)\}_{i=1}^n$ to train a reward model $R_\theta$. For the adversarial platform, it also trains $\widetilde{R}_{\widetilde{\theta}}$ using perturbed preference data $\widetilde{\mathcal{D}}(\boldsymbol{\delta})$. The reward models $R$ and $\widetilde{R}$ may or may not have the same architecture, we discuss details in Appendix C.1.

**Iterative retraining process.** At each iteration, the generative model produces 50,000 data samples, of which 25,000 are selected (after curation by the reward model) for retraining the next-generation model. In Appendix C, we provide details on the process of (adversarial) data curation and their interaction with the generative model training. The complete process is presented in Algorithm 2.

### 5.1. Evolution of model under adversarial data curation

First, we examine the impact of adversarial data curation on the self-consuming generative model $p_t$ and the respective reward $\mathbb{E}_{p_t}[r(x)]$. Specifically, we employ both the gradient-based attack (Algorithm 1) and random attack (i.e., flip

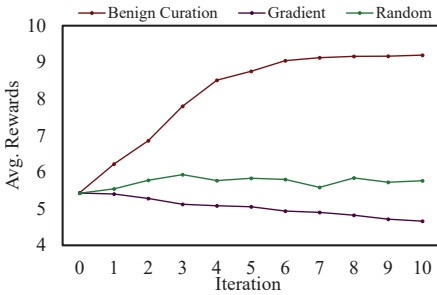

*Figure 4.* Empirical estimate of $\mathbb{E}_{p_t}[r(x)]$ from model-generated samples over iterations on CIFAR-10: it increases (resp. decreases) over iterations under benign (resp. adversarial) data curation.

preference labels uniformly at random) algorithms to target 20% of data pairs.

Fig. 2 shows the evolution of the proportion of each class generated during the iterative retraining process on CIFAR-10 and it demonstrates the impact of three types of data curation on self-consuming models: benign curation, adversarial curation via gradient-based attacks, and adversarial curation via random attacks. Without adversarially curated data (left), the generative model gradually aligns with human preferences, producing an increasing number of samples from class 0 (airplane), the most preferred class in CIFAR-10. This observation is consistent with Ferbach et al. (2024). Under random attacks, while the model does not align as well with user preferences as in benign curation, it still generates a reasonable proportion of samples from the most favored classes, 0 (airplane) and 1 (automobile). In contrast, with our proposed attack algorithm, the model becomes highly misaligned with user preferences, generating samples predominantly from the least favored classes: 7 (horse), 8 (ship), and 9 (frog).

Fig. 4 shows the empirical estimate of $\mathbb{E}_{p_t}[r(x)]$ for generated samples over iterations on CIFAR-10, with observations consistent with Lemma 3.3. Under benign curation, the expected reward steadily increases as the model aligns with user preferences. In contrast, adversarial curation using the gradient-based attack continuously lowers the reward, indicating a significant deviation from the optimal reward distribution. Random attacks initially lead to a slight increase in rewards, suggesting some robustness. However, as attacks progress, the average reward fluctuates due to the inherent randomness of this method. Indeed, the outcomes for random attacks vary significantly across different runs of experiments, and we provide additional examples in Appendix C.1.

We extend our experiments to CIFAR-100, which contains a larger number of classes. Fig. 3 illustrates the different behaviors under benign curation and adversarial curation using the gradient-based attack. With benign curation, the model progressively generates more samples from user-preferred classes, reflecting an increasing alignment with user prefer-

*Table 1.* Effectiveness of one-round attack under different methods on the same CIFAR-10 dataset: The results are empirical estimates of $\mathbb{E}_{p_1}[r(x)]$ for generated samples at $t = 1$; method with a lower reward is more effective.

| METHOD | BENIGN | GRADIENT #1 | GRADIENT #2 |
|--------|--------|-------------|-------------|
| AVG. R | 6.3606 | 5.6460 | 5.5959 |
| METHOD | PARETO | R-BASED #1 | R-BASED #2 |
| AVG. R | 5.4740 | 5.5982 | 5.5612 |

ences over time. In contrast, under adversarial curation, the model generates more samples from less preferred classes (i.e., classes 60 to 99). The average rewards shown on the right further highlight this difference: benign curation leads to a steady increase in expected rewards, whereas adversarial curation causes a continuous decline, indicating growing misalignment between the model and user preferences.

We also conducted iterative model retraining on a mixture of adversarially curated synthetic and real data on CIFAR-10, where adversarial data generated through gradient-based attacks is combined with real data at varying proportions. Fig. 5 shows the class proportions of the generated samples over iterations. The results demonstrate that incorporating real data helps align the distribution of generated samples with the real data distribution. However, it does not steer the model toward the user-preferred distribution. This suggests that adding a limited amount of real data is insufficient and fails to defend against adversarially curated data.

To examine model quality, we present synthetic images generated by self-consuming models under both adversarial and benign data curation in Fig. 7 (Appendix C.1). The results show that the image quality does not significantly degrade during the iterative retraining process.

### 5.2. Effectiveness of attack methods

To evaluate the effectiveness of various attack algorithms, we applied different attack methods to the same CIFAR-10 dataset, flipping 20% of the data pairs. Table 1 summarizes the empirical estimates of $\mathbb{E}_{p_1}[r(x)]$ for generated samples at $t = 1$. Comparisons include benign curation (BENIGN), gradient-based attacks that target platform and adversarial platform using different (GRADIENT #1) or identical (GRADIENT #2) reward models, reward-based heuristic methods with $f(x, z) = |R_\theta(x) - R_\theta(z)|$ (R-BASED #1) or $\max\{|R_\theta(x)|, |R_\theta(z)|\}$ (R-BASED #2), and multi-objective heuristic method (PARETO), where we select the solution from the Pareto front with the lowest sum of the two metrics. We also show the class proportions of generated samples over iterations in Fig. 8 (Appendix C.1).

Overall, the results demonstrate the effectiveness of all attack methods in this experimental scenario. Despite employing different reward model architectures, gradient-based

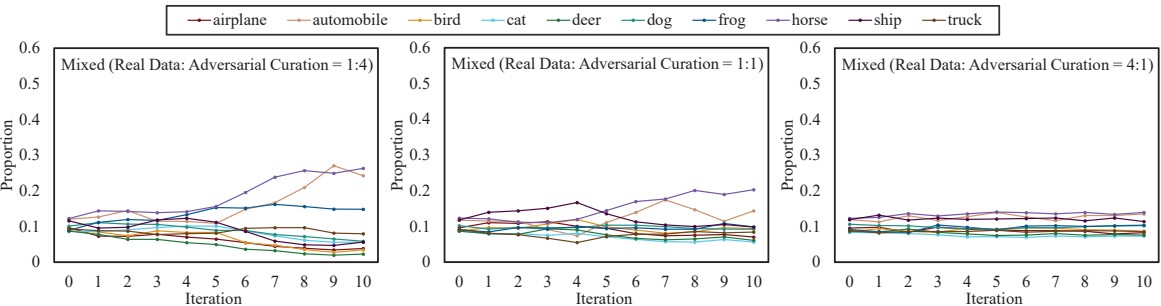

*Figure 5.* The proportion of each class generated by the self-consumption model retrained with a mix of adversarially curated synthetic and real CIFAR-10 data. It shows that adding real data only helps the model align with the real data distribution $p_{\text{data}}$ but does not defend against adversarial data curation.

attack methods consistently maintained high efficacy. In contrast, the relatively weaker performance of heuristic methods is reflected in their higher average rewards. As shown in Fig. 8, this is primarily due to heuristic methods failing to account for all classes comprehensively, resulting in limited effects on certain classes. The multi-objective heuristic method performs best by exploring a broader range of potential solutions, yielding the lowest average reward. However, this performance comes at the cost of significantly higher computational time and resource requirements.

It is important to note that the effectiveness of these attack methods may depend heavily on the specific context, and no method can be considered "optimal". Future research could focus on developing solutions that are more general and universally applicable.

## 6. Discussion

In this section, we discuss the robustness of existing defense strategies and analyze the practical challenges of defending against our proposed attack. We also discuss key assumptions made in our analysis and identify limitations that may affect applicability in real-world scenarios.

**Defense.** A common strategy proposed in prior work to stabilize the self-consuming retraining loop of generative models is to regularly inject real data during training (Ale-mohammad et al., 2024a; Bertrand et al., 2024). However, as shown in the experiments in Fig. 5, adding real data only partially mitigates adversarial effects by driving the model closer to the true data distribution $p_{\text{data}}$. It does not effectively prevent model misalignment under targeted adversarial curation attacks.

Although outlier detection may assist in identifying and filtering adversarially curated samples, they can inadvertently remove genuine preferences. When users are heterogeneous and come from multiple groups, removing genuine preferences from minority groups may potentially introduce

biases. Additionally, our attack algorithm already considers such defense mechanisms: when formulating in Eq. (10), we impose a penalty term $\text{dist}(R_\theta, \widetilde{R}_{\widetilde{\theta}})$ to prevent the adversarial behavior from being easily detected as anomalous.

**Limitations.** Our theoretical analysis assumes that each model update converges to the global optimum of the training objective. While our experiments validate the effectiveness of the proposed attack under this assumption, such convergence may not always hold in practice due to optimization noise, local minima, or limited training budgets.

Additionally, our experiments rely on a known success ratio $\kappa$ to control the adversarial curation. In real-world scenarios, however, the attacker may not have direct access to or precise knowledge of the effective success rate, which could affect the practical impact of the attack.

## 7. Conclusion

This paper examines the evolution of self-consuming generative models under adversarially curated data. We theoretically analyze the impact of adversarial data curation on these models and identify conditions for the (in)stability of the iterative retraining process. Building on theoretical insights, we develop attack algorithms that effectively disrupt model training and prevent alignment with user preferences. The findings highlight the potential vulnerability of self-consuming generative models to adversarial data curation, suggesting that developing effective defense mechanisms could be a promising direction for future work.

## Acknowledgement

This material is based upon work supported by the U.S. National Science Foundation under award IIS2202699 and IIS-2416895, by OSU President's Research Excellence Accelerator Grant, and grants from the Ohio State University's Translational Data Analytics Institute and College of Engineering Strategic Research Initiative.

## Impact Statement

This paper advances the field of Machine Learning by examining the long-term impact of the self-consuming retraining loop on generative models, as well as the role of (adversarial) data curation in this process. There are many potential societal consequences of our work, none of which we feel must be specifically highlighted here.

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

# Appendix

# A. Related work

## A.1. Self-consuming generative models

Generative models have demonstrated remarkable success in synthesizing high-quality data. Recent research has focused on the challenges faced by generative models trained iteratively on their own synthetic data. Despite slight differences in the definition of collapse, Alemohammad et al. (2024a); Shumailov et al. (2024); Gerstgrasser et al. (2024) all confirm that the model collapses when trained exclusively on synthetic datasets, leading to a degradation in quality or diversity over successive generations. Bertrand et al. (2024); Taori & Hashimoto (2023) theorizes that this collapse arises due to the nature of the iterative training process. Another potential consequence of this iterative training is bias amplification, where the generative model amplifies specific features while neglecting other equally important data characteristics (Chen et al., 2024; Taori & Hashimoto, 2023; Wyllie et al., 2024).

There are two main solutions to this problem. One effective approach to stabilizing generative models is to introduce real data into the training process at each iteration. Alemohammad et al. (2024a) provided empirical evidence that injecting fresh real data mitigates model collapse. Bertrand et al. (2024) further substantiated this observation with theoretical proofs, demonstrating that maintaining a sufficient proportion of real data ensures the stability of iterative training. Gerstgrasser et al. (2024) leverages cumulative data, where previously generated samples are stored and reused alongside new real data to stabilize the performance of the generative model. This approach aligns with practical data accumulation strategies. An alternative strategy is to introduce corrective mechanisms to prevent model collapse. Alemohammad et al. (2024b) proposed a self-improvement framework for generative diffusion models (SIMS), which mitigates model collapse by employing a negative guidance mechanism during the generation process. Gillman et al. (2025) introduced a self-correcting generative model training loop, where synthetic data undergoes transformation through expert-informed correction functions before being reintroduced into the training set. This method significantly enhances stability by ensuring that synthetic samples retain high fidelity and do not reinforce existing biases.

Recent studies have also examined the role of human feedback in iterative retraining. Ferbach et al. (2024) explored how user-curated synthetic samples can implicitly optimize a model's reward function, aligning generative outputs with human preferences. While preference alignment improves the user experience, it can also introduce systematic biases that may be exploited by adversaries. This potential issue forms the basis for the exploration in our work.

## A.2. Data poisoning attack

Data poisoning attack is an attack that disrupts model learning by modifying training data (Vorobeychik & Kantarcioglu, 2018). Two major methods of this attack are the injection of poison data and label-flipping attacks.

One common data poisoning technique involves adding maliciously produced samples to the training dataset. In generative models, Jiang et al. (2024) demonstrates that poisoning even a small fraction of the training data can significantly alter the model's output, such as in text summarization or text completion tasks. Carlini et al. (2024) shows how adversaries can inject poisoned examples into datasets at minimal cost by exploiting web-based data collection mechanisms, which is not merely a theoretical concern but a practical threat. Furthermore, Baumgärtner et al. (2024) explored the vulnerability of large language model (LLM) fine-tuning to poisoning attacks. In this scenario, an adversary can skillfully modify a small portion of the training data, injecting undetectable biases into the generated text, potentially leading to misinformation or biased outputs.

The label flipping attack modifies the underlying true labels assigned to a subset of training samples. This attack is particularly effective in classification and regression tasks (Biggio et al., 2011; Liu et al., 2017; Paudice et al., 2019; Suya et al., 2023), where incorrect labels can mislead the model's learning process. In recommendation systems, Zhang et al. (2020) proposed an adversarial reinforcement learning approach that strategically flips labels to manipulate item rankings. Similarly, in Reinforcement Learning with Human Feedback (RLHF) training, preference poisoning attacks introduce incorrect label flips in reward datasets, causing reinforcement learning models to produce biased responses (Wu et al., 2025). Several studies have explored defenses against label flipping attacks. Traditional methods include outlier detection techniques (Zeng et al., 2023), which identify and discard suspicious training samples. Paudice et al. (2019) introduced a robust filtering approach that leverages bilevel optimization to identify and remove mislabeled samples from poisoned datasets, providing an effective solution against label flipping attacks.

Unlike previous research that focuses on flipping labels to promote or demote a specific target, our approach emphasizes perturbing the entire alignment process. Notably, our attacker does not require access to data collection pipelines or backend systems; instead, they can act entirely through public feedback mechanisms, such as voting or ranking systems. This makes our approach more practical and harder to detect, as it unfolds gradually over time without direct data manipulation. Whereas traditional poisoning attacks often aim to induce outright model failure, our objective is more subtle: gradually misaligning the model from genuine user preferences over time in a self-consuming training loop. In competitive settings, such gradual misalignment can be highly damaging while remaining difficult to trace.

## B. Proofs

### B.1. Proof of Lemma 3.1

**Lemma B.1.** *Let $p_{t+1}$ be defined as in Equation* (4). *If we assume that $\mathbb{E}_{z \sim p_t}[e^{r(z)}] < \infty$, then we have for all $x \in \mathbb{R}^d$,*

$$p_{t+1}(x) \xrightarrow{K \to \infty} p_t(x) \left[ (1 - \phi_t) \frac{e^{r(x)}}{\mathbb{E}_{z \sim p_t}\left[e^{r(z)}\right]} + \phi_t \frac{e^{\widetilde{r}_t(x)}}{\mathbb{E}_{z \sim p_t}\left[e^{\widetilde{r}_t(z)}\right]} \right] \tag{12}$$

*Proof.* Consider the limit of $K \to \infty$. By minimization of the cross-entropy, we know that for any distribution $q$, $\arg\max_p \mathbb{E}_{x \sim q}[\log(p(x))] = q$.

So, sample $K$ samples from $p(t)$ independently and identically distributed, then sample $i_K$ with the following probability:

$$\mathbb{P}(i_K = i | x_1, ..., x_K) = \phi_t \frac{e^{\widetilde{r}_t(x_i)}}{\sum_{j=1}^{K} e^{\widetilde{r}_t(x_j)}} + (1 - \phi_t) \frac{e^{r(x_i)}}{\sum_{j=1}^{K} e^{r(x_j)}}$$

Noting that the events $\{i_K = i\}_{i=1}^{K}$ are disjoint, the resulting density can be written:

$$
\begin{aligned}
p_{t+1}(x) &= \sum_{i=1}^{K} \int_{y_j, j \neq i} p_t(y_1, \cdots, y_{i-1}, x, y_{i+1}, \cdots, y_K) \mathbb{P}(i_K = i \mid x, y_j, j \neq i) \prod_{j \neq i} dy_j \\
&= K \int_{y_1, \cdots, y_{K-1}} p_t(y_1, \cdots, y_{K-1}, x) \mathbb{P}(i_K = K \mid y_1, \cdots, y_{K-1}, x) \, dy_1 \cdots dy_{K-1} \\
&= p_t(x) K (1 - \phi_t) \int_{y_1, \cdots, y_{K-1}} \frac{e^{r(x)}}{e^{r(x)} + \sum_{i=1}^{K-1} e^{r(y_i)}} p_t(y_1) \cdots p_t(y_{K-1}) \, dy_1 \cdots dy_{K-1} \\
&\quad + p_t(x) K \phi_t \int_{y_1, \cdots, y_{K-1}} \frac{e^{\widetilde{r}_t(x)}}{e^{\widetilde{r}_t(x)} + \sum_{i=1}^{K-1} e^{\widetilde{r}_t(y_i)}} p_t(y_1) \cdots p_t(y_{K-1}) \, dy_1 \cdots dy_{K-1} \\
&= p_t(x) \cdot \left[ (1 - \phi_t) H_{p_t}^K(x) + \phi_t \widetilde{H}_{p_t}^K(x) \right]
\end{aligned}
$$

where

$$H_{p_t}^K(x) = \int_{y_1, \cdots, y_{K-1}} \frac{e^{r(x)}}{\frac{e^{r(x)}}{K} + \frac{K-1}{K} \frac{\sum_{i=1}^{K-1} e^{r(y_i)}}{K-1}} p_t(y_1) \cdots p_t(y_{K-1}) \, dy_1 \cdots dy_{K-1}$$

$$\widetilde{H}_{p_t}^K(x) = \int_{y_1, \cdots, y_{K-1}} \frac{e^{\widetilde{r}_t(x)}}{\frac{e^{\widetilde{r}_t(x)}}{K} + \frac{K-1}{K} \frac{\sum_{i=1}^{K-1} e^{\widetilde{r}_t(y_i)}}{K-1}} p_t(y_1) \cdots p_t(y_{K-1}) \, dy_1 \cdots dy_{K-1}$$

When $K \to \infty$:

$$H_{p_t}^K(x) \xrightarrow{K \to \infty} \frac{e^{r(x)}}{\mathbb{E}_{z \sim p_t}\left[e^{r(z)}\right]}$$

$$\widetilde{H}_{p_t}^K(x) \xrightarrow{K \to \infty} \frac{e^{\widetilde{r}_t(x)}}{\mathbb{E}_{z \sim p_t}\left[e^{\widetilde{r}_t(z)}\right]}$$

So,

$$p_{t+1}(x) \xrightarrow{K \to \infty} p_t(x) \left[ (1 - \phi_t) \frac{e^{r(x)}}{\mathbb{E}_{z \sim p_t} \left[ e^{r(z)} \right]} + \phi_t \frac{e^{\widetilde{r}_t}(x)}{\mathbb{E}_{z \sim p_t} \left[ e^{\widetilde{r}_t(z)} \right]} \right]$$

$\square$

## B.2. Additional lemma: the reward expectation is not increasing

Without assuming that the reward is bounded, we can show using Jensen inequality that:

**Lemma B.2.** *When performing K-wise filtering, $\forall t \geq 0$, the expected reward:*

$$\mathbb{E}_{p_{t+1}} \geq \mathbb{E}_{x \sim p_t} \left[ e^{r(x)} \right] + \phi_t \, \mathrm{Cov}_{x \sim p_t} \left[ e^{r(x)}, e^{\widetilde{r}_t(x)} \right] \tag{13}$$

*Proof.* Suppose $Z = \frac{K-1}{K} \frac{\sum_{i=1}^{K-1} e^{r(z_i)}}{K-1}$, by Jensen inequality, for any $x$ ($\frac{a}{b+x}$ is convex):

$$H_{p_t}^K(x) = \mathbb{E}_Z \left[ \frac{e^{r(x)}}{\frac{e^{r(x)}}{K} + Z} \right] \geq \frac{e^{r(x)}}{\frac{e^{r(x)}}{K} + \mathbb{E}[Z]} = \frac{e^{r(x)}}{\frac{e^{r(x)}}{K} + \frac{K-1}{K} \mathbb{E}_{p_t} \left[ e^{r(x)} \right]}$$

$$\widetilde{H}_{p_t}^K(x) = \mathbb{E}_Z \left[ \frac{e^{\widetilde{r}(x)}}{\frac{e^{\widetilde{r}(x)}}{K} + \widetilde{z}} \right] \geq \frac{e^{\widetilde{r}x)}}{\frac{e^{\widetilde{r}(x)}}{K} + \mathbb{E}[Z]} = \frac{e^{\widetilde{r}(x)}}{\frac{e^{\widetilde{r}(x)}}{K} + \frac{K-1}{K} \mathbb{E}_{p_t} \left[ e^{\widetilde{r}(x)} \right]}$$

So:

$$\mathbb{E}_{p_{t+1}} \left[ e^{r(x)} \right] = \int e^{r(x)} p_t(x) \cdot \left[ (1 - \phi_t) H_{p_t}^K(x) + \phi_t \widetilde{H}_{p_t}^K(x) \right] dx$$

$$= (1 - \phi_t) \int e^{r(x)} p_t(x) H_{p_t}^K(x) dx + \phi_t \int e^{r(x)} p_t(x) \widetilde{H}_{p_t}^K(x) dx$$

$$\geq (1 - \phi_t) \int p_t(x) \frac{e^{2r(x)}}{\frac{e^{r(x)}}{K} + \frac{K-1}{K} \mathbb{E}_{z \sim p_t} \left[ e^{r(z)} \right]} dx + \phi_t \int p_t(x) \frac{e^r(x) e^{\widetilde{r}_t(x)}}{\frac{e^{\widetilde{r}_t(x)}}{K} + \frac{K-1}{K} \mathbb{E}_{\widetilde{z} \sim p_t} \left[ e^{\widetilde{r}_t(\widetilde{z})} \right]} dx$$

$$(\mathbb{E}_{x \sim p_t}[f(x)] = \int p_t(x) f(x) dx)$$

$$= (1 - \phi_t) \mathbb{E}_{x \sim p_t} \frac{e^{2r(x)}}{\frac{e^{r(x)}}{K} + \frac{K-1}{K} \mathbb{E}_{z \sim p_t} \left[ e^{r(z)} \right]} + \phi_t \mathbb{E}_{x \sim p_t} \frac{e^r(x) e^{\widetilde{r}_t(x)}}{\frac{e^{\widetilde{r}_t(x)}}{K} + \frac{K-1}{K} \mathbb{E}_{\widetilde{z} \sim p_t} \left[ e^{\widetilde{r}_t(\widetilde{z})} \right]}$$

(Jensen's inequality)

$$\geq (1 - \phi_t) \frac{\mathbb{E}_{x \sim p_t} \left[ e^{r(x)} \right]^2}{\frac{\mathbb{E}_{x \sim p_t} \left[ e^{r(x)} \right]}{K} + \frac{K-1}{K} \mathbb{E}_{z \sim p_t} \left[ e^{r(z)} \right]} + \phi_t \frac{\mathbb{E}_{x \sim p_t} \left[ e^r(x) e^{\widetilde{r}_t(x)} \right]}{\frac{\mathbb{E}_{x \sim p_t} \left[ e^{\widetilde{r}_t(x)} \right]}{K} + \frac{K-1}{K} \mathbb{E}_{z \sim p_t} \left[ e^{\widetilde{r}_t(z)} \right]}$$

$$= (1 - \phi_t) \mathbb{E}_{x \sim p_t} \left[ e^{r(x)} \right] + \phi_t \frac{\mathbb{E}_{x \sim p_t} \left[ e^{r(x)} e^{\widetilde{r}_t(x)} \right]}{\mathbb{E}_{x \sim p_t} \left[ e^{\widetilde{r}_t(x)} \right]}$$

$$= (1 - \phi_t) \mathbb{E}_{x \sim p_t} \left[ e^{r(x)} \right] + \phi_t \, \mathrm{Cov}_{x \sim p_t} \left[ e^{r(x)}, e^{\widetilde{r}_t(x)} \right] + \phi_t \mathbb{E}_{x \sim p_t} \left[ e^{r(x)} \right]$$

$$= \mathbb{E}_{x \sim p_t} \left[ e^{r(x)} \right] + \phi_t \, \mathrm{Cov}_{x \sim p_t} \left[ e^{r(x)}, e^{\widetilde{r}_t(x)} \right]$$

That is:

$$\mathbb{E}_{p_{t+1}} \geq \mathbb{E}_{x \sim p_t} \left[ e^{r(x)} \right] + \phi_t \, \mathrm{Cov}_{x \sim p_t} \left[ e^{r(x)}, e^{\widetilde{r}_t(x)} \right]$$

$\square$

## B.3. Proof of Lemma 3.3

**Lemma B.3.** *Let $p_{t+1}$ the distribution induced from a discrete choice model on in Eq.(4). Suppose Assumption 3.2 holds, then the expected reward,*

$$
\begin{aligned}
\mathbb{E}_{p_{t+1}}\left[e^{r(x)}\right] &\geq \mathbb{E}_{p_t}\left[e^{r(x)}\right] + (1-\phi_t)\frac{(K-1)}{K}\frac{\operatorname{Var}_{p_t}\left[e^{r(x)}\right]}{e^{r_{t,\max}}} + \phi_t\frac{(K-1)}{K}\frac{\operatorname{Cov}_{p_t}\left[e^{r(x)}, e^{\widetilde{r}_t(x)}\right]}{e^{\widetilde{r}_{t,\min}}} \\
\mathbb{E}_{p_{t+1}}\left[e^{r(x)}\right] &\leq \mathbb{E}_{p_t}\left[e^{r(x)}\right] + (1-\phi_t)\frac{(K-1)}{K}\frac{\operatorname{Var}_{p_t}\left[e^{r(x)}\right]}{e^{r_{t,\min}}} + \phi_t\frac{(K-1)}{K}\frac{\operatorname{Cov}_{p_t}\left[e^{r(x)}, e^{\widetilde{r}_t(x)}\right]}{e^{\widetilde{r}_{t,\max}}}
\end{aligned}
\tag{14}
$$

*Proof.*

$$
\begin{aligned}
K\mathbb{E}_{p_{t+1}}\left[e^{r(x)}\right] =& K\int_{x_1,\cdots,x_K}(1-\phi_t)\frac{e^{2r(x_1)}+\cdots+e^{2r(x_K)}}{e^{r(x_1)}+\cdots+e^{r(x_K)}} + \phi_t\frac{e^{\widetilde{r}_t(x_1)}e^{r(x_1)}+\cdots+e^{\widetilde{r}_t(x_K)}e^{r(x_K)}}{e^{\widetilde{r}_t(x_1)}+\cdots+e^{\widetilde{r}_t(x_K)}}\prod_{k=1}^{K}p_t(x_k)\,dx_k \\
=&(1-\phi_t)\int_{x_1,\cdots,x_K}K\frac{e^{2r(x_1)}+\cdots+e^{2r(x_K)}}{e^{r(x_1)}+\cdots+e^{r(x_K)}}\prod_{k=1}^{K}p_t(x_k)\,dx_k \\
&+\phi_t\int_{x_1,\cdots,x_K}K\frac{e^{\widetilde{r}_t(x_1)}e^{r(x_1)}+\cdots+e^{\widetilde{r}_t(x_K)}e^{r(x_K)}}{e^{\widetilde{r}_t(x_1)}+\cdots+e^{\widetilde{r}_t(x_K)}}\prod_{k=1}^{K}p_t(x_k)\,dx_k \\
=&(1-\phi_t)\int_{x_1,\ldots,x_K}\sum_{j=1}^{K}\left[e^{r(x_j)}\frac{e^{r(x_1)}+\cdots+e^{r(x_K)}}{e^{r(x_1)}+\cdots+e^{r(x_K)}} + e^{r(x_j)}\frac{(K-1)e^{r(x_j)}-\sum_{i\neq j}e^{r(x_i)}}{e^{r(x_1)}+\cdots+e^{r(x_K)}}\right] \\
&\prod_{k=1}^{K}p_t(x_k)\,dx_k \\
&+\phi_t\int_{x_1,\ldots,x_K}\sum_{j=1}^{K}\left[e^{r(x_j)}\frac{e^{\widetilde{r}_t(x_1)}+\cdots+e^{\widetilde{r}_t(x_K)}}{e^{\widetilde{r}_t(x_1)}+\cdots+e^{\widetilde{r}_t(x_K)}} + e^{r(x_j)}\frac{(K-1)e^{\widetilde{r}_t(x_j)}-\sum_{i\neq j}e^{\widetilde{r}_t(x_i)}}{e^{\widetilde{r}_t(x_1)}+\cdots+e^{\widetilde{r}_t(x_K)}}\right] \\
&\prod_{k=1}^{K}p_t(x_k)\,dx_k \\
=&(1-\phi_t)\left[K\mathbb{E}_{p_t}\left[e^{r(x)}\right] + \int_{x_1,\ldots,x_K}\frac{\sum_{i<j}\left(e^{r(x_i)}-e^{r(x_j)}\right)^2}{e^{r(x_1)}+\cdots+e^{r(x_K)}}\prod_{k=1}^{K}p_t(x_k)\,dx_k\right] \\
&+\phi_t\left[K\mathbb{E}_{p_t}\left[e^{r(x)}\right] + \int_{x_1,\ldots,x_K}\frac{\sum_{i<j}\left(e^{r(x_i)}-e^{r(x_j)}\right)\left(e^{\widetilde{r}_t(x_i)}-e^{\widetilde{r}_t(x_j)}\right)}{e^{\widetilde{r}_t(x_1)}+\cdots+e^{\widetilde{r}_t(x_K)}}\prod_{k=1}^{K}p_t(x_k)\,dx_k\right]
\end{aligned}
$$

Suppose

$$
A = \int_{x_1,\ldots,x_K}\frac{\sum_{i<j}\left(e^{r(x_i)}-e^{r(x_j)}\right)^2}{e^{r(x_1)}+\cdots+e^{r(x_K)}}\prod_{k=1}^{K}p_t(x_k)\,dx_k
$$

and

$$
B = \int_{x_1,\ldots,x_K}\frac{\sum_{i<j}\left(e^{r(x_i)}-e^{r(x_j)}\right)\left(e^{\widetilde{r}_t(x_i)}-e^{\widetilde{r}_t(x_j)}\right)}{e^{\widetilde{r}_t(x_1)}+\cdots+e^{\widetilde{r}_t(x_K)}}\prod_{k=1}^{K}p_t(x_k)\,dx_k
$$

Because $\operatorname{Var}_{p_t}\left[e^{r(x)}\right] \geq 0$, $A \geq \sum_{i<j}\frac{2\operatorname{Var}_{p_t}\left[e^{r(x)}\right]}{Ke^{r_{t,\max}}}$

When $\operatorname{Cov}_{p_t}\left[e^{r(x)}, e^{\widetilde{r}_t(x)}\right] > 0$, $B \geq \sum_{i<j}\frac{2\operatorname{Cov}_{p_t}\left[e^{r(x)}, e^{\widetilde{r}_t(x)}\right]}{Ke^{\widetilde{r}_{t,\max}}} \geq 0$; when $\operatorname{Cov}_{p_t}\left[e^{r(x)}, e^{\widetilde{r}_t(x)}\right] < 0$, $B \geq \sum_{i<j}\frac{2\operatorname{Cov}_{p_t}\left[e^{r(x)}, e^{\widetilde{r}_t(x)}\right]}{Ke^{\widetilde{r}_{t,\min}}} < 0$, so $B \geq \sum_{i<j}\frac{2\operatorname{Cov}_{p_t}\left[e^{r(x)}, e^{\widetilde{r}_t(x)}\right]}{Ke^{\widetilde{r}_{t,\min}}}$

Then, we have:

$$
\begin{aligned}
K\mathbb{E}_{p_{t+1}}\left[e^{r(x)}\right] \geq & (1-\phi_t)\left[K\mathbb{E}_{p_t}\left[e^{r(x)}\right] + \sum_{i<j}\frac{2\operatorname{Var}_{p_t}\left[e^{r(x)}\right]}{Ke^{r_{t,\max}}}\right] + \phi_t\left[K\mathbb{E}_{p_t}\left[e^{r(x)}\right] + \sum_{i<j}\frac{2\operatorname{Cov}_{p_t}\left[e^{r(x)},e^{\widetilde{r}_t(x)}\right]}{Ke^{\widetilde{r}_{t,\min}}}\right] \\
\geq & (1-\phi_t)\left[K\mathbb{E}_{p_t}\left[e^{r(x)}\right] + \frac{K(K-1)}{2}\frac{2\operatorname{Var}_{p_t}\left[e^{r(x)}\right]}{Ke^{r_{t,\max}}}\right] \\
& + \phi_t\left[K\mathbb{E}_{p_t}\left[e^{r(x)}\right] + \frac{K(K-1)}{2}\frac{2\operatorname{Cov}_{p_t}\left[e^{r(x)},e^{\widetilde{r}_t(x)}\right]}{Ke^{\widetilde{r}_{t,\min}}}\right] \\
= & K\mathbb{E}_{p_t}\left[e^{r(x)}\right] + (1-\phi_t)\frac{(K-1)\operatorname{Var}_{p_t}\left[e^{r(x)}\right]}{e^{r_{t,\max}}} + \phi_t\frac{(K-1)\operatorname{Cov}_{p_t}\left[e^{r(x)},e^{\widetilde{r}_t(x)}\right]}{e^{\widetilde{r}_{t,\min}}}
\end{aligned}
$$

which means:

$$
\mathbb{E}_{p_{t+1}}\left[e^{r(x)}\right] \geq \mathbb{E}_{p_t}\left[e^{r(x)}\right] + (1-\phi_t)\frac{(K-1)}{K}\frac{\operatorname{Var}_{p_t}\left[e^{r(x)}\right]}{e^{r_{t,\max}}} + \phi_t\frac{(K-1)}{K}\frac{\operatorname{Cov}_{p_t}\left[e^{r(x)},e^{\widetilde{r}_t(x)}\right]}{e^{\widetilde{r}_{t,\min}}}
$$

Similarly, we have: $A \leq \sum_{i<j}\frac{2\operatorname{Var}_{p_t}\left[e^{r(x)}\right]}{Ke^{r_{t,\min}}}$ and $B \leq \sum_{i<j}\frac{2\operatorname{Cov}_{p_t}\left[e^{r(x)},e^{\widetilde{r}_t(x)}\right]}{Ke^{\widetilde{r}_{t,\max}}}$

So:

$$
\mathbb{E}_{p_{t+1}}\left[e^{r(x)}\right] \leq \mathbb{E}_{p_t}\left[e^{r(x)}\right] + (1-\phi_t)\frac{(K-1)}{K}\frac{\operatorname{Var}_{p_t}\left[e^{r(x)}\right]}{e^{r_{t,\min}}} + \phi_t\frac{(K-1)}{K}\frac{\operatorname{Cov}_{p_t}\left[e^{r(x)},e^{\widetilde{r}_t(x)}\right]}{e^{\widetilde{r}_{t,\max}}}
$$

$\square$

We also prove the rationality of the upper bound. That is, the upper bound is always greater than 0.

*Proof.* Suppose $r_{t,\min} \leq r(x) \leq r_{t,\max}$ and $\widetilde{r}_{t,\min} \leq \widetilde{r}(x) \leq \widetilde{r}_{t,\max}$.

Then, we have $e^{r_{t,\min}} \leq \mathbb{E}_{p_t}\left[e^{r(x)}\right] \leq e^{r_{t,\max}}$ and $e^{2r_{t,\min}} \leq \mathbb{E}_{p_t}\left[e^{2r(x)}\right] \leq e^{2r_{t,\max}}$.

Because $0 \leq \operatorname{Var}_{p_t}\left[e^{r(x)}\right] \leq \mathbb{E}_{p_t}\left[e^{2r(x)}\right]$, then $0 \leq \operatorname{Var}_{p_t}\left[e^{r(x)}\right] \leq e^{2r_{t,\max}}$.

Because $\operatorname{Cov}_{p_t}\left[e^{r(x)},e^{\widetilde{r}_t(x)}\right] = \mathbb{E}_{p_t}\left[e^{r(x)}e^{\widetilde{r}_t(x)}\right] - \mathbb{E}_{p_t}\left[e^{r(x)}\right]\mathbb{E}_{p_t}\left[e^{\widetilde{r}_t(x)}\right]$, so $e^{r_{t,\min}+\widetilde{r}_{t,\min}} - e^{r_{t,\max}+\widetilde{r}_{t,\max}} \leq \operatorname{Cov}_{p_t}\left[e^{r(x)},e^{\widetilde{r}_t(x)}\right] \leq e^{r_{t,\max}+\widetilde{r}_{t,\max}} - e^{r_{t,\min}+\widetilde{r}_{t,\min}}$.

Then

$$
\begin{aligned}
& \mathbb{E}_{p_t}\left[e^{r(x)}\right] + (1-\phi_t)\frac{(K-1)}{K}\frac{\operatorname{Var}_{p_t}\left[e^{r(x)}\right]}{e^{r_{t,\min}}} + \phi_t\frac{(K-1)}{K}\frac{\operatorname{Cov}_{p_t}\left[e^{r(x)},e^{\widetilde{r}_t(x)}\right]}{e^{\widetilde{r}_{t,\max}}} \\
\geq & e^{r_{t,\min}} + (1-\phi_t)\frac{(K-1)}{K}\frac{0}{e^{r_{t,\min}}} + \phi_t\frac{(K-1)}{K}\frac{e^{r_{t,\min}+\widetilde{r}_{t,\min}} - e^{r_{t,\max}+\widetilde{r}_{t,\max}}}{e^{\widetilde{r}_{t,\max}}} \\
= & e^{r_{t,\min}} + \phi_t\frac{(K-1)}{K}\frac{e^{r_{t,\min}+\widetilde{r}_{t,\min}} - e^{r_{t,\max}+\widetilde{r}_{t,\max}}}{e^{\widetilde{r}_{t,\max}}} \\
= & e^{r_{t,\min}} + \phi_t\frac{(K-1)}{K}(e^{r_{t,\min}+\widetilde{r}_{t,\min}-\widetilde{r}_{t,\max}} - e^{r_{t,\max}+\widetilde{r}_{t,\max}-\widetilde{r}_{t,\max}}) \\
= & e^{r_{t,\min}} + \phi_t\frac{(K-1)}{K}(e^{r_{t,\min}+\widetilde{r}_{t,\min}-\widetilde{r}_{t,\max}} - e^{r_{t,\max}}) \\
= & e^{r_{t,\min}} + e^{r_t t,\min}\phi_t\frac{(K-1)}{K}(e^{\widetilde{r}_{t,\min}-\widetilde{r}_{t,\max}} - e^{r_{t,\max}-r_{t,\min}})
\end{aligned}
$$

When $\operatorname{Var}_{p_t}\left[e^{r(x)}\right] = 0$, $e^{r_{t,\min}} = e^{r_{t,\max}}$

So:

$$
\begin{aligned}
& \mathbb{E}_{p_t}\left[e^{r(x)}\right] + (1-\phi_t)\frac{(K-1)}{K}\frac{\operatorname{Var}_{p_t}\left[e^{r(x)}\right]}{e^{r_{t,\min}}} + \phi_t\frac{(K-1)}{K}\frac{\operatorname{Cov}_{p_t}\left[e^{r(x)},e^{\widetilde{r}_t(x)}\right]}{e^{\widetilde{r}_{t,\max}}} \\
\geq & e^{r_{t,\min}} + e^{r_{t,\min}}\phi_t\frac{(K-1)}{K}(e^{\widetilde{r}_{t,\min}-\widetilde{r}_{t,\max}} - 1)
\end{aligned}
$$

Because $0 < e^{\widetilde{r}_{t,\min} - \widetilde{r}_{t,\max}} \leq 1$, $-1 < e^{\widetilde{r}_{t,\min} - \widetilde{r}_{t,\max}} - 1 \leq 0$

Because $0 \leq \phi_t \frac{(K-1)}{K} \leq 1$, then $-e^{r_{t,\min}} < e^{r_{t,\min}}(e^{\widetilde{r}_{t,\min} - \widetilde{r}_{t,\max}} - 1) \leq 0$

That is

$$e^{r_{t,\min}} + e^{r_{t,\min}}\phi_t \frac{(K-1)}{K}(e^{\widetilde{r}_{t,\min} - \widetilde{r}_{t,\max}} - 1) > 0$$

which means

$$\mathbb{E}_{p_t}\left[e^{r(x)}\right] + (1 - \phi_t)\frac{(K-1)}{K}\frac{\mathrm{Var}_{p_t}\left[e^{r(x)}\right]}{e^{r_{t,\min}}} + \phi_t\frac{(K-1)}{K}\frac{\mathrm{Cov}_{p_t}\left[e^{r(x)}, e^{\widetilde{r}_t(x)}\right]}{e^{\widetilde{r}_{t,\max}}} > 0$$

$\square$

## B.4. Proof of Lemma 3.4

**Lemma B.4.** *Let $p_{t+1}$ be defined as in Eq. (3). And suppose $\mathrm{Cov}_{\min} = \min_{i\in\{0,1,\dots,t\}}\mathrm{Cov}_{p_i}\left[e^{r(x)}, e^{\widetilde{r}_i(x)}\right]$ and $\phi_t = \phi_{t-1} = \cdots = \phi_1 = \phi_\star$. With $p_0 = p_{data}$, for $\forall t > 1$:*

$$\mathbb{E}_{p_{t+1}}\left[e^{r(x)}\right] \geq \mathbb{E}_{p_{data}}\left[e^{r(x)}\right] + \phi_\star(1+\lambda)\left(1 - \left(\frac{\lambda}{1+\lambda}\right)^t\right)\mathrm{Cov}_{\min} \tag{15}$$

*When $t \to \infty$,*

$$\mathbb{E}_{p_{t+1}}\left[e^{r(x)}\right] \geq \mathbb{E}_{p_{data}}\left[e^{r(x)}\right] + \phi_\star(\lambda+1)\mathrm{Cov}_{\min} \tag{16}$$

*Proof.* According to the Lemma B.2

$$\mathbb{E}_{p_1} \geq \mathbb{E}_{p_0}\left[e^{r(x)}\right] + \phi_\star\mathrm{Cov}_{p_0}\left[e^{r(x)}, e^{\widetilde{r}_0(x)}\right]$$

Then

$$\mathbb{E}_{p_2}\left[e^{r(x)}\right] \geq \frac{1}{1+\lambda}\mathbb{E}_{p_{data}}\left[e^{r(x)}\right] + \frac{\lambda}{1+\lambda}\left(\mathbb{E}_{p_0}\left[e^{r(x)}\right] + \phi_\star\mathrm{Cov}_{p_0}\left[e^{r(x)}, e^{\widetilde{r}_0(x)}\right]\right)$$

With $p_0 = p_{data}$

$$\mathbb{E}_{p_2}\left[e^{r(x)}\right] \geq \mathbb{E}_{p_{data}}\left[e^{r(x)}\right] + \frac{\lambda}{1+\lambda}\phi_\star\mathrm{Cov}_{p_0}\left[e^{r(x)}, e^{\widetilde{r}_0(x)}\right]$$

Use recursion:

$$\mathbb{E}_{p_{t+1}}\left[e^{r(x)}\right] \geq \mathbb{E}_{p_{data}}\left[e^{r(x)}\right] + \phi_\star\left[\left(\frac{\lambda}{1+\lambda}\right)\mathrm{Cov}_{p_t}\left[e^{r(x)}, e^{\widetilde{r}_t(x)}\right] + \left(\frac{\lambda}{1+\lambda}\right)^2\mathrm{Cov}_{p_{t-1}}\left[e^{r(x)}, e^{\widetilde{r}_{t-1}(x)}\right]\right.$$
$$\left. + \cdots + \left(\frac{\lambda}{1+\lambda}\right)^t\mathrm{Cov}_{p_0}\left[e^{r(x)}, e^{\widetilde{r}_0(x)}\right]\right]$$

Suppose

$$\mathrm{Cov}_{\min} = \mathrm{Cov}_{\min}\left[e^{r(x)}, e^{\widetilde{r}_m(x)}\right] = \min\left\{\mathrm{Cov}_{p_t}\left[e^{r(x)}, e^{\widetilde{r}_t(x)}\right], \mathrm{Cov}_{p_{t-1}}\left[e^{r(x)}, e^{\widetilde{r}_{t-1}(x)}\right], \dots, \mathrm{Cov}_{p_0}\left[e^{r(x)}, e^{\widetilde{r}_0(x)}\right]\right\}$$

Then,

$$\mathbb{E}_{p_{t+1}}\left[e^{r(x)}\right] \geq \mathbb{E}_{p_{data}}\left[e^{r(x)}\right] + \phi_\star\left[\left(\frac{\lambda}{1+\lambda}\right)\mathrm{Cov}_{\min}\left[e^{r(x)}, e^{\widetilde{r}_m(x)}\right] + \left(\frac{\lambda}{1+\lambda}\right)^2\mathrm{Cov}_{\min}\left[e^{r(x)}, e^{\widetilde{r}_m(x)}\right]\right.$$
$$\left. + \cdots + \left(\frac{\lambda}{1+\lambda}\right)^t\mathrm{Cov}_{\min}\left[e^{r(x)}, e^{\widetilde{r}_m(x)}\right]\right]$$
$$= \mathbb{E}_{p_{data}}\left[e^{r(x)}\right] + \phi_\star(1+\lambda)\left(1 - \left(\frac{\lambda}{1+\lambda}\right)^t\right)\mathrm{Cov}_{\min}$$

When $t \to \infty$,

$$\mathbb{E}_{p_{t+1}}\left[e^{r(x)}\right] \geq \mathbb{E}_{p_{data}}\left[e^{r(x)}\right] + \phi_\star(\lambda+1)\mathrm{Cov}_{\min}$$

$\square$

---

**Algorithm 2** Iterative retraining with adversarially curated data

---

**Input:** Real data $\mathcal{D}_{data} = \{d_i\}_{i=1}^N$, user reward function $r$, learning procedure of generative model $\mathcal{G}$, learning procedure of reward model $\mathcal{R}$, attack algorithms $\mathcal{A}$

**Param:** Rate of attack $\kappa$, proportion of data $\beta$, proportion of filtered data $\lambda$

$p_0 = \mathcal{G}(\mathcal{D}_{data})$

**for** $t = 1$ **to** $T$ **do**

$\quad \mathcal{D}_{gen} = \{\widetilde{d}_i\}_{i=1}^N$, where $\widetilde{d}_1, \ldots, \widetilde{d}_N \sim p_{t-1}$

$$\mathcal{D}_{=}\{(x_i, y_i, o_i)\}_{i=1}^n, \text{ where } x_i, y_i \sim p_{t-1}, o_i = \begin{cases} 1 & \text{if } r(x_i) < r(y_i), \\ 0.5 & \text{if } r(x_i) = r(_i), \\ 0 & \text{if } r(x_i) > r(y_i). \end{cases}$$

$\quad \widetilde{\mathcal{D}} = \mathcal{A}(D, \kappa)$

$\quad R_\theta = \mathcal{R}(\mathcal{D}), \widetilde{R}_{\widetilde{\theta}} = \mathcal{R}(\widetilde{\mathcal{D}})$

$\quad \mathcal{D}_t = \text{Curate}(\mathcal{D}_{gen}, R_\theta / \widetilde{R}_{\widetilde{\theta}}, \beta)$ {Using $R_\theta$ or $\widetilde{R}_{\widetilde{\theta}}$ curated $\beta N$ data on the generated sample set $\mathcal{D}_{gen}$}

$\quad p_t = \mathcal{G}(\mathcal{D}_{data} \cup \lambda \mathcal{D}_t)$

**end for**

---

## C. Additional experiments

Algorithm 2 describes the process of iterative retraining on the adversarial curation synthetic dataset and the real dataset.

### C.1. Additional experiments on CIRFAR-10 datasets

In this section, we show some additional experiments on the CIRFAR-10 dataset.

**Settings.** The settings in this section are the same as those described in Section 5. For the reward $R$ and $\tilde{R}$ may share the same architecture or differ. If they have the same architecture, we use pretrained VGG11 as the feature extractor and a linear layer containing 10 neurons for both $R$ and $\tilde{R}$. If different architectures are used, $R$ employs a pretrained VGG11 as the feature extractor, followed by three linear layers with 128, 64, and 10 neurons, respectively.

**Two other random attacks experiments.** As we mentioned in Section 5.1, the results of the random attacks are not exactly similar. We show the proportions of each class and the average reward values with iteration number for two other independent experiments in Fig. 6. These results further highlight the variability of random attacks. RANDOM #1 shows a case where the average reward continues to increase in the later stages of retraining, indicating partial alignment with user preferences. RANDOM #2 shows a gradual decrease in the average reward throughout the process, reflecting a more persistent misalignment.

**Additional results of adversarial data curation experiments.** As we mentioned in Section 5.1, we show the samples generated during the retraining process with different curation in Fig 7. It can be seen that the images are not distorted to the point of being unrecognizable, but there is a noticeable change in the diversity of the generated samples. In the benign

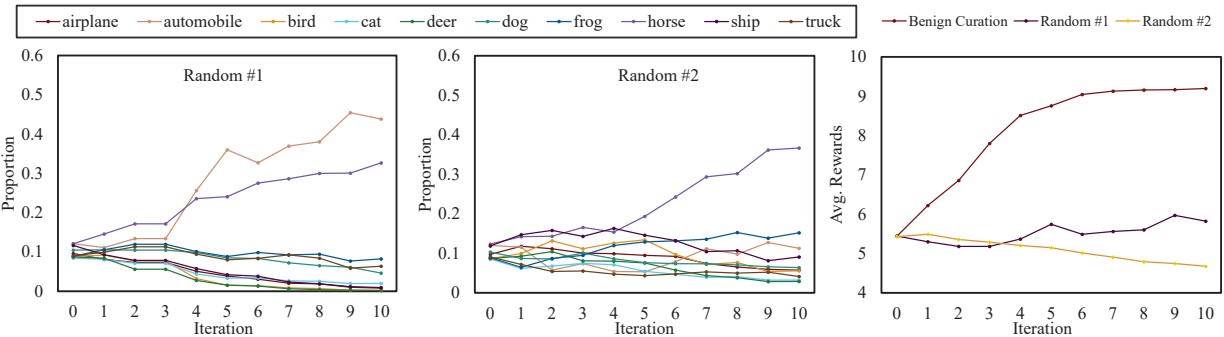

*Figure 6.* Iterative retraining of the self-consumption model on the CIFAR-10 dataset with two independent experiments employing a randomized adversarial curation (20% random): (1) Left side: proportion of each class for the two independent runs. (2) Right side: average reward values for the two independent runs and benign curation

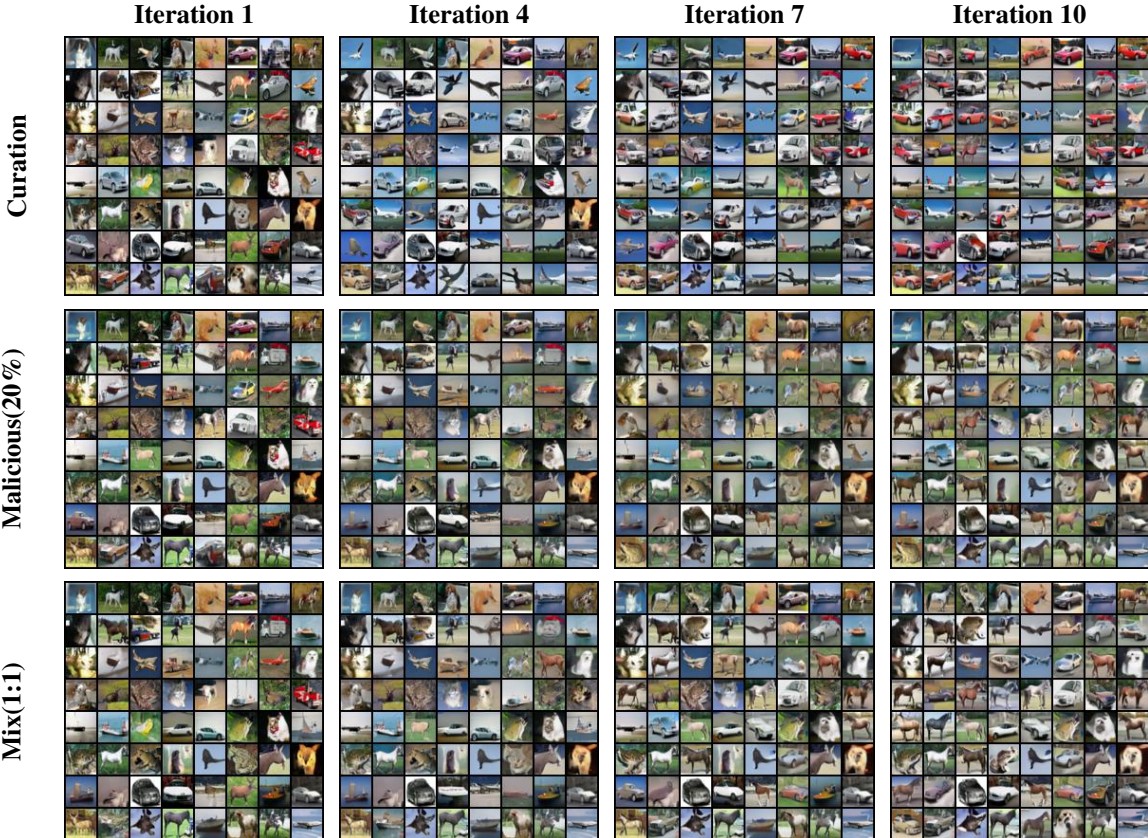

*Figure 7.* Samples generated by self-consumption model with different curation on CIFAR-10 dataset:(1) Top: bengiun curation, filtered by $r(x)$. (2) Middle: adversarial curation with 20% malicious data injected by gradient algorithm. (3) Bottom: mixed dataset created by combining real data with adversarially curated synthetic data (1:1).

curation setting, the model generates a large proportion of images from user-preferred classes (airplane and automobile). In contrast, the malicious curatios are dominated by less favored classes (horse and ship), indicating a significant misalignment with intended user preferences.

**Additional results of the attack algorithm experiments.** As we mentioned in Section 5.2, in the experiment we recorded the proportions of each classification, and the results are shown in Fig 8. Comparisons include benign curation (BENIGN), gradient-based attacks that target platform and adversarial platform using different reward models (GRADIENT #1) or identical reward models(GRADIENT #2), reward-based heuristic methods with $f(x, z) = |R_\theta(x) - R_\theta(z)|$ (HEURISTIC #1) or $\max\{|R_\theta(x)|, |R_\theta(z)|\}$ (HEURISTIC #2), and multi-objective heuristic method (PARETO).

An interesting phenomenon is that although the heuristic has a lower average reward value, it has a significantly higher proportion of automobile (the user's preferred category) when curated. This may be due to the fact that the heuristics are not sufficiently global.

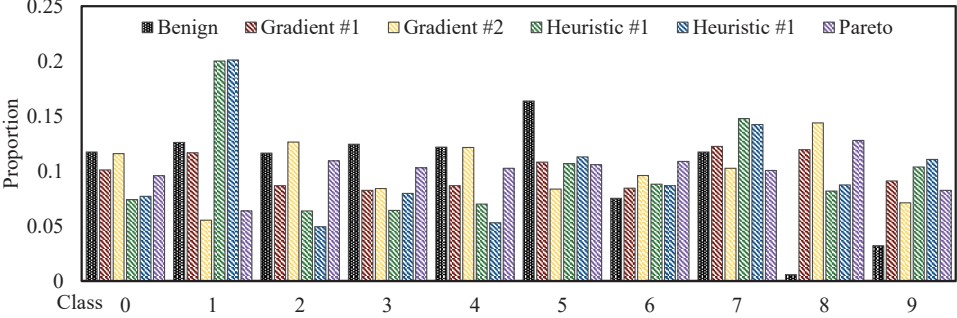

*Figure 8.* The proportion of each class on at $t = 1$ under different attack algorithms on the same CIFAR-10 dataset.

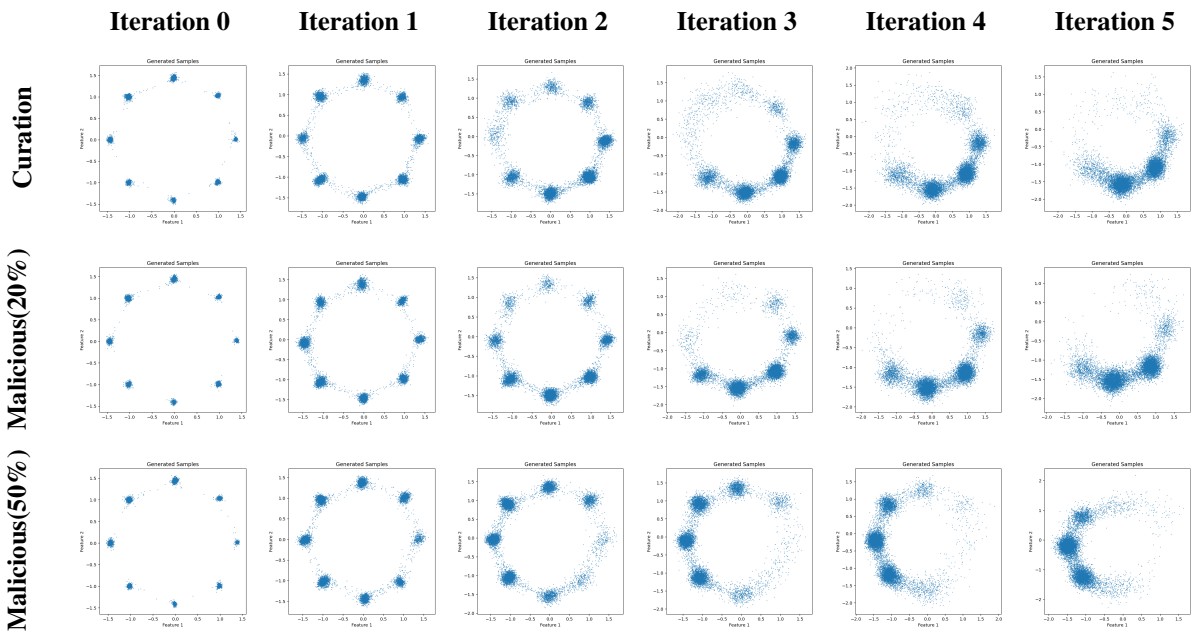

*Figure 9.* Samples generated by self-consumption model with different curation on synthetic Gaussian dataset:(1) Top: bengiun curation, filtered by $r(x)$. (2) Middle: adversarial curation with 20% malicious data injected by gradient algorithm. (3) Bottom: adversarial curation with 50% malicious data injected by gradient algorithm (sever attack).

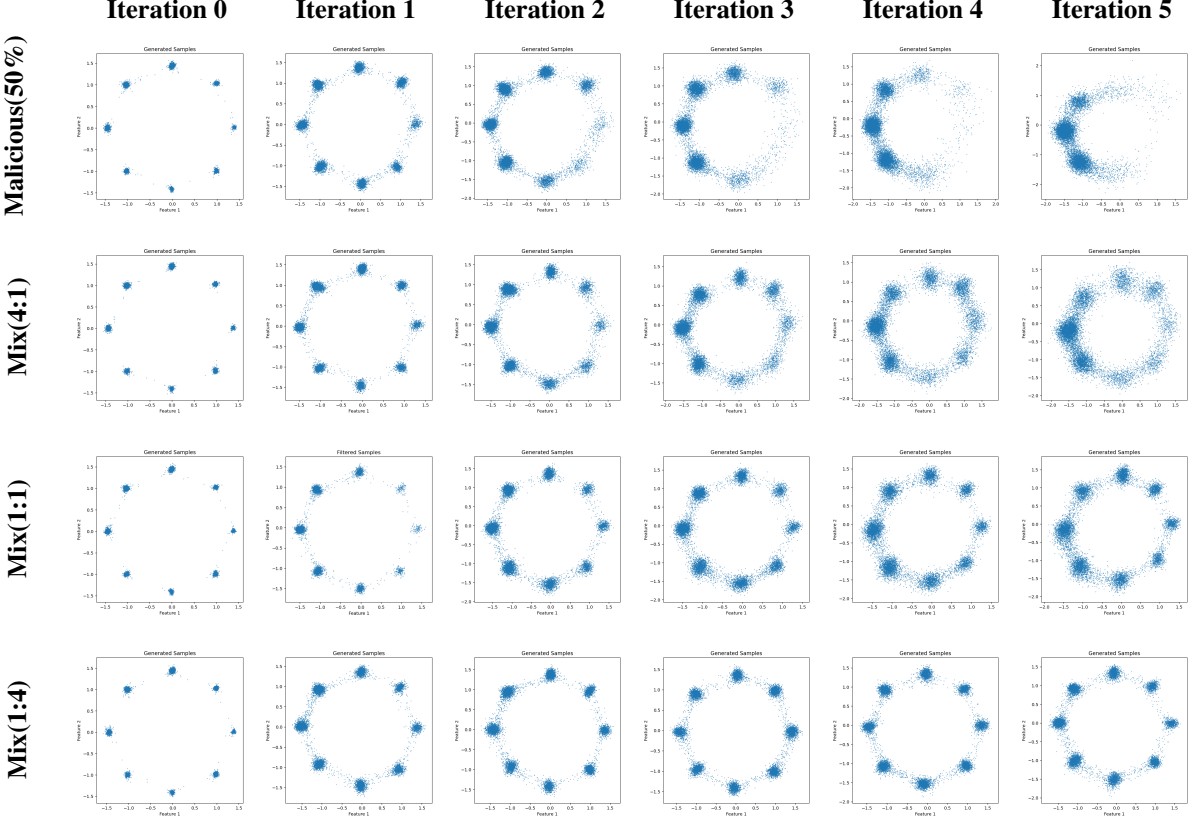

*Figure 10.* Samples generated by self-consumption model on different mixed Gaussian dataset:(1) Top: adversarially curated synthetic dataset with 50% malicious data injected by gradient algorithm (sever attack). (2) Second to Bottom: mixed dataset created by combining real data with adversarially curated synthetic data in different proportions (real data : adversarially curated synthetic data)

### C.2. Experiments on Gaussian datasets

This section shows the setting and results of the experiments on the synthetic dataset.

**Dataset.** The synthetic dataset we generated is a two-dimensional dataset following an 8-mode Gaussian mixture model. Specifically, we define eight mode centers that are uniformly distributed on a circle of radius 2. The coordinates of these centers are given by:

$$\mu_t = 2 \times \left( \cos(\frac{t\pi}{4}), \sin(\frac{t\pi}{4}) \right), \quad t = 0, 1, 2, \ldots, 7. \tag{17}$$

Each data point is independently and uniformly sampled from the 8 mode points, and isotropic Gaussian noise with a mean of 0 and a standard deviation of 0.02 is added:

$$x = \mu_t + \epsilon, \quad \epsilon \sim \mathcal{N}(0, 0.02^2 I_2). \tag{18}$$

**Settings.** For the $r(x) := -\gamma \max\{0, \|x - \mu_*\| - \tau\}$, we designate $\mu_* = (2, 0)$ (which is the first center) , $\tau = 3$ and $\gamma = -10$. And the reward model we used consists of two fully connected linear layers with 2 neuron in first layer and 64 neurons in the second layer. In each iteration, the generative model produces $10,000$ random samples, from which $5,000$ samples are filtered for next retraining. The samples generated in each iteration are plotted on two-dimensional coordinates.

**Adversarial curation.** We explored the long-term performance on purely synthetic data with adversarial curation. The results are shown as Fig.9, which shows three different curation: benign curation, adversarial curation using gradient descent algorithms attacking 20% and 50% data pairwise datasets, respectively.

**Mixed data.** We also explored the long-term performance of adversarial curation on mixed data. The resutls are shown as Fig.10. Under severe attacks, adding a large amount of real data can align it to the real data distribution, but not to the user preference distribution.

