# OpenReview forum: "Self-Consuming Generative Models with Adversarially Curated Data"
_ICML.cc/2025/Conference — ICML 2025 poster_

### Official Review · Reviewer_VYFS · 2025-03-10

**Overall Recommendation:** 3

**Summary:**

This paper investigates the effects of adversarially curated data on generative models trained iteratively on synthetic data—referred to as "self-consuming loops." The authors theoretically and experimentally analyze how generative models behave under conditions of noisy and maliciously curated data. They propose algorithms to strategically disrupt a competitor's model training by adversarially manipulating data curation processes. Key findings include the identification of conditions under which models either remain robust and converge to optimize user preferences or become misaligned due to adversarial manipulation.

**Claims And Evidence:**

The claims of model robustness and vulnerability under adversarial conditions are supported convincingly through both theoretical analysis (Lemma 3.3 and Lemma 3.4) and experimental validation. The effectiveness of proposed attack algorithms is demonstrated clearly via experiments.

**Essential References Not Discussed:**

There are no essential missing references.

**Experimental Designs Or Analyses:**

The experimental designs are sound and systematically validate theoretical insights. The comparison between benign, random, and adversarially curated datasets clearly demonstrates the impact of adversarial attacks.

**Methods And Evaluation Criteria:**

The methods and evaluation criteria (theoretical analyses and benchmark datasets such as CIFAR-10 and synthetic Gaussian datasets) are highly appropriate and relevant for the studied problem.

**Other Comments Or Suggestions:**

NA

**Other Strengths And Weaknesses:**

Strengths:
- Strong theoretical grounding clearly defining model behavior under adversarial data curation.
- Novel adversarial attack algorithms effectively disrupting generative model alignment.
- Extensive experimental validation demonstrating clear practical implications.

Weaknesses:
- The gradient-based attack methods are computationally expensive, potentially limiting their practical deployment.
- Experiments are primarily demonstrated on CIFAR-10 and synthetic data; further validation on larger-scale or more diverse datasets would strengthen the claims.

**Questions For Authors:**

Have the authors considered strategies to mitigate or detect such adversarial curation attacks in practical scenarios?

**Relation To Broader Scientific Literature:**

The paper effectively situates itself within current research, highlighting its novelty as the first exploration of adversarial data curation in the context of self-consuming generative loops. It clearly distinguishes its contributions from prior studies (e.g., Ferbach et al., 2024; Wu et al., 2024), emphasizing the novel consideration of adversarial user curation in iterative training loops.

**Theoretical Claims:**

The methods and evaluation criteria (theoretical analyses and benchmark datasets such as CIFAR-10 and synthetic Gaussian datasets) are highly appropriate and relevant for the studied problem.

---

> ### Author Rebuttal · Authors · 2025-04-01
>
> We thank the reviewer for their time and encouraging feedback, especially for recognizing the novelty and effectiveness of our empirical validation. We address the reviewer’s concerns below:
>
> > “The gradient-based attack methods are computationally expensive, potentially limiting their practical deployment.”
>
> We agree with the reviewer that gradient-based adversarial curation methods can be computationally expensive, especially in large-scale or real-time settings. However, we emphasize that we were aware of this limitation and have addressed it in the paper by proposing alternative attack strategies that do not rely on gradient computations  (See Section 4.2. Heuristic methods). While each approach has its own limitations, all were shown to be effective in the experiments.
>
> Our work is the first to study curation attacks in the self-consuming generative retraining process. We believe that designing more efficient or scalable attack algorithms is an important direction for future research.
>
> > “Experiments are primarily demonstrated on CIFAR-10 and synthetic data; further validation on larger-scale or more diverse datasets would strengthen the claims. Further validation on larger-scale or more diverse datasets would strengthen the claims.”
>
> As noted by the reviewer, we currently demonstrate results on CIFAR-10 and synthetic data. We chose these datasets to align with prior works [1, 2], which evaluate the self-consuming retraining loop under benign data curation. We fully agree with the reviewer that evaluating on larger or more diverse datasets would strengthen our findings. However, due to time and compute constraints, we were unable to complete experiments on very large datasets during the rebuttal period.
>
> Instead, we have extended our experiments to CIFAR-100, a more diverse dataset with 100 classes. The results can be found at [3], which validate the theorem and demonstrate the effectiveness of the proposed algorithm. The observations are consistent with our findings: benign curation gradually steers the class distribution toward user preferences, leading to progressively increasing reward scores. In contrast, adversarial curation with the gradient-based attack disrupts this alignment, depresses reward growth, and drives the model away from the user preferences.
>
> > "Have the authors considered strategies to mitigate or detect such adversarial curation attacks in practical scenarios?"
>
> While this work focuses on analyzing the vulnerability of self-consuming retraining loops to adversarial data curation, we have indeed considered potential mitigation strategies:
>
> - Adding real data: This is a common strategy used in prior works to stabilize the self-consuming retraining loop of generative models [4, 5]. We have evaluated this method in our paper. As shown in the experiments (Fig. 3), adding real data only partially mitigates adversarial effects by driving the model closer to the true data distribution $p\_{data}$. However, it does not fully prevent misalignment.
>
> - Anomaly detection methods: Although approaches like outlier detection may help identify and eliminate adversarially curated samples, they can inadvertently remove genuine preferences. When users are heterogeneous and come from multiple groups, removing genuine preferences from minority groups may potentially introduce biases. Additionally, our attack algorithm already considers such defense mechanisms: when formulating optimization (10), we impose a penalty term $\text{dist}(R\_{\theta},\widetilde{R}\_{\widetilde{\theta}})$ to prevent the adversarial behavior from being easily detected as anomalous (see line 261, left column).
>
> We believe that designing effective defense mechanisms is an important direction for future research. We would be happy to include the above discussion in the revised version of the paper.
>
> [1] Ferbach, D., Bertrand, Q., Bose,A.J., and Gidel, G. Self consuming generative models with curated data provably optimize human preferences, 2024.
>
> [2] Bertrand, Q., Bose, A.J., Duplessis, A., et al. On the stability of iterative retraining of generative models on their own data, 2024.
>
> [3] https://anonymous.4open.science/r/Anonymize_ICML2025-4CC5/CIFAR100.jpg
>
> [4] Alemohammad, S., Casco-Rodriguez, J., Luzi, L., et al. Self-consuming generative models go mad, 2023.
>
> [5] Bertrand, Q., Bose, A.J., Duplessis, A., et al. On the stability of iterative retraining of generative models on their own data, 2024.

---

### Official Review · Reviewer_vxoQ · 2025-03-11

**Overall Recommendation:** 4

**Summary:**

This paper investigates a novel adversarial model where the data curation process for generating training data for iterative models is adversarially manipulated. The authors show theoretically that the effectiveness of such adversarial manipulations have is tied to the covariance between the unmanipulated and manipulated distributions. The theory is extended to account for the case where some fraction of the initial training distribution is always used. Leveraging theoretical insights, the authors propose two methods for how such adversarial manipulation could be performed and empirically demonstrate their efficacy.

## update after rebuttal
I appreciate the additional clarifications from the authors. I have no serious concerns (mostly just on nomenclature), and I stand by my original positive review.

**Claims And Evidence:**

The theoretical claims are proven in the appendix, and the empirical claims are supported by experiments under two settings: Gaussian Mixture and CIFAR-10.

**Essential References Not Discussed:**

Not to my knowledge.

**Experimental Designs Or Analyses:**

The experimental design is fine, I have one minor concern about how the $\kappa$ parameter is treated, please see “Questions for Authors”.

**Methods And Evaluation Criteria:**

Yes, the evaluations are sensible.

**Other Comments Or Suggestions:**

Please see my comments in “Questions for Authors”

**Other Strengths And Weaknesses:**

One minor note is that the theory seems to assume that training converges to the global optimum. It is unclear how crucial that condition is to the overall theoretical analysis.

**Questions For Authors:**

- It looks like Lemma 3.3 doesn’t necessarily suggest that the distribution will converge to the optimal value, since it is possible for an increasing sequence to converge to a suboptimal value. Is there an additional piece here that ensures optimal convergence?

- On page 5, it is stated that $\kappa$ represents the success rate of perturbing the data on the target platform. Doesn’t that mean that you would need to account for the fact that the attacker does not have control over which manipulated samples are curated? It seems as though it is treated more like a budget in the experiments.

**Relation To Broader Scientific Literature:**

This work is extends the work of Ferbach et al., which investigates the dynamics of iterative retraining with human curated data, to the adversarial case. More generally, it introduces a new adversarial model.

**Theoretical Claims:**

I did not throughly review the proofs attached in the appendix.

---

> ### Author Rebuttal · Authors · 2025-04-01
>
> We thank the reviewer for the detailed feedback and positive assessment of our work. We appreciate the recognition of our theoretical analysis, experimental design, and the contribution of introducing a new adversarial model in the context of iterative retraining. We address the reviewer's concerns below:
>
> > "One minor note is that the theory seems to assume that training converges to the global optimum. It is unclear how crucial that condition is to the overall theoretical analysis."
>
> The reviewer is correct. Our theoretical analysis assumes that each model update converges to the global optimum of the training objective. (i.e., $p\_t$ maximizes data log-likelihood). This is a standard simplification in prior works [1, 2] to facilitate the analytical analysis.
>
> However, we agree that this assumption may not always hold in practice. Nevertheless, we emphasize that our main theoretical insights remain valid as long as each model update sufficiently approximates the optimum. In fact, our empirical results show that the observed behaviors persist even when training is noisy or approximate. We appreciate the reviewer for highlighting this point, and we will clarify this assumption and its implications in the revised version.
>
> > “Lemma 3.3 doesn’t necessarily suggest that the distribution will converge to the optimal value.”
>
> Yes, Lemma 3.3 does not claim that the iteratively retrained model converges to the optimum; instead, it only characterizes the relationship between the expected reward $\mathbb{E}\_{p\_{t}}\left[e^{r(x)}\right]$ over two consecutive time steps. As discussed in lines 172-185 (left column), whether the expected reward converges to the maximum depends on the covariance term $\operatorname{Cov}\_{p\_{t}}\left[e^{r(x)},e^{\widetilde{r}\_{t}(x)}\right]$. When the covariance is positive at every step, the expected reward increases and the variance decreases, indicating convergence toward the optimal distribution (which aligns with the findings in [1]). However, if the covariance becomes negative, the expected reward may oscillate and deviate from the maximum value. Indeed, our attack algorithm is developed based on these theoretical insights: when misaligning the model from human preferences, optimization (10) aims to flip human preference labels such that the covariance is as negative as possible.
>
> > About $\kappa$: the success rate
>
> The reviewer’s understanding is correct. In real-world settings, the attacker typically does not have full control over which adversarial samples are ultimately curated on the target platform. This is exactly why we avoid referring to $\kappa$ as *budget*, and instead interpret it as the success rate of perturbing the data on the target platform.
>
> In the experiments, we treat $\kappa$ as a controllable parameter to analyze how different attack intensities impact model alignment. We will add this clarification in the revised manuscript.
>
> [1] Ferbach, D., Bertrand, Q., Bose,A.J., and Gidel, G. Self consuming generative models with curated data provably optimize human preferences, 2024.
>
> [2] Bertrand, Q., Bose, A.J., Duplessis, A., et al. On the stability of iterative retraining of generative models on their own data, 2024.

---

### Official Review · Reviewer_YfNj · 2025-03-13

**Overall Recommendation:** 3

**Summary:**

This work proposes a method for adversarial attack defense when training generative models. Experimental results on synthetic and real datasets show the effectiveness.

**Claims And Evidence:**

All claims have support in the paper.

**Essential References Not Discussed:**

No.

**Experimental Designs Or Analyses:**

The utilized datasets are very simple. I suggest utilizing datasets that contain more categories, such as ImageNet, for better evaluation. The introduced baseline, DDPM, might not represent all generative models, while methods based on GANs and VAEs also attracted much research attention recently.

**Methods And Evaluation Criteria:**

The method makes sense for current generative models.

**Other Comments Or Suggestions:**

As mentioned, I suggest considering more experiments.

**Other Strengths And Weaknesses:**

None.

**Questions For Authors:**

I have no more questions.

**Relation To Broader Scientific Literature:**

It is a new method for generative models' adversarial attack field.

**Theoretical Claims:**

I do not check the proofs carefully but take a quick look. I do not notice obvious errors.

---

> ### Author Rebuttal · Authors · 2025-04-01
>
> We thank the reviewer for the constructive comments and for recognizing our work as "a new method in generative models' adversarial attack field." We now address the reviewer's concerns regarding the experiment:
>
> > On dataset diversity and model generality
>
> Our experimental setup follows prior work [1, 2], which also uses synthetic and CIFAR-10 datasets to analyze self-consuming retraining. In addition to reproducing the expected reward maximization under benign curation, our experiments also demonstrate the impact of adversarial curation, compare different attack strategies, and show that the model outputs remain visually plausible under attack.
>
> We acknowledge that including a broader range of generative models, such as GANs and VAEs, would make our evaluation more comprehensive and convincing. While our experiments present results using DDPM, we adopt a theoretical framework that does not depend on specific architectures. Our analysis applies to any likelihood-based model, including VAEs. Extending this work to GANs would require reformulating the training dynamics, which we leave for future work.
>
> Similarly, we agree that evaluating on more complex datasets (such as ImageNet) would strengthen the empirical validation of our method. However, due to time and compute constraints, we were unable to complete these experiments during the rebuttal period.
>
> Instead, we have extended our experiments to CIFAR-100, a more diverse dataset with 100 classes. The results can be found at [3], which validate the theorem and demonstrate the effectiveness of the proposed algorithm. The observations are consistent with our findings: benign curation gradually steers the class distribution toward user preferences, leading to progressively increasing reward scores. In contrast, adversarial curation with the gradient-based attack disrupts this alignment, depresses reward growth, and drives the model away from the user preferences.
>
> [1] Ferbach, D., Bertrand, Q., Bose,A.J., and Gidel, G. Self consuming generative models with curated data provably optimize human preferences, 2024.
>
> [2] Bertrand, Q., Bose, A.J., Duplessis, A., et al. On the stability of iterative retraining of generative models on their own data, 2024.
>
> [3] https://anonymous.4open.science/r/Anonymize_ICML2025-4CC5/CIFAR100.jpg

---

### Official Review · Reviewer_A97s · 2025-03-13

**Overall Recommendation:** 4

**Summary:**

The paper studies the problem of iterative retraining of generative models on their own synthetic data, in the specific setting where synthetic data have been adversarially curated, e.g., a concurrent platform gives random or adversarial feedback when to vote for their favorite image on MidJourney. In this setting, authors theoretically show that the iteratively trained generative model learns the curation mixture probability. In addition they provide experiments to illustrate that iterative retraining on adversarially curated data do not maximize the initial, non-adversarial reward.

**Claims And Evidence:**

Yes.

**Essential References Not Discussed:**

NA

**Experimental Designs Or Analyses:**

Yes.

**Methods And Evaluation Criteria:**

Yes.

**Other Comments Or Suggestions:**

NA

**Other Strengths And Weaknesses:**

Other Strengths And Weaknesses:
Strength:
The paper is very well written and very clear. The proofs are correct and well supported by the experiments.

Weaknesses:
My main concern is the **plausibility of the proposed scenario** and the impact of the reach conclusions: do we know how frequently theses curation attacks are likely to happen?
Even if these curation happen, the conclusion feels a bit "obvious": "if someone is injecting bad, then iterative retraining of generative models will fail maximizing the reward".
It seems that these questions are especially relevant since the setting, proofs, and experiments are mostly incremental with respect to Ferbach 2024.

I would be eager to learn about academic references to be back the proposed attack scenario, do the authors think about Carlini 2024 implementation of attacks?

CARLINI, Nicholas, JAGIELSKI, Matthew, CHOQUETTE-CHOO, Christopher A., et al. Poisoning web-scale training datasets is practical. In : 2024 IEEE Symposium on Security and Privacy (SP). IEEE, 2024. p. 407-425.

**Questions For Authors:**

NA

**Relation To Broader Scientific Literature:**

The broader literature is correctly addressed.

**Theoretical Claims:**

Results seem correct. I foresee no issue. I check the proofs of Lemmas 3.1 - 3.3.

---

> ### Author Rebuttal · Authors · 2025-04-01
>
> We thank the reviewer for the thoughtful and encouraging feedback, especially their positive comments on the theoretical analysis, experimental design, and for describing our paper as “well written and very clear”. We provide clarifications to address the main concerns:
>
> > "Do we know how frequently theses curation attacks are likely to happen?"
>
> In practice, the frequency of curation attacks depends on how often a model is iteratively retrained using human-curated feedback. Precisely quantifying this frequency requires collecting real-world data from the target platform, which is an interesting direction for future research. Nevertheless, we emphasize that the proposed scenario is realistic: as generative models increasingly rely on human feedback for training and updates, it opens an opportunity for curation attacks.
>
> For example, InstructGPT [1] and LLaMA-2-Chat [2] are fine-tuned using curated human preferences; Pick-a-Pic [3] and Rich Feedback for Text-to-Image [4] demonstrate how human judgments can guide and refine image generation. If users can influence model updates through their preferences, then malicious feedback, which is curated by adversarial users such as those employed by competing platforms, may steer the model away from genuine user intent. Our work addresses a timely and critical question: under what conditions does adversarial curation impact the iterative training of generative models intended to align with benign user preferences?
>
> > "Do the authors think about Carlini 2024 implementation of attacks?"
>
> We thank the reviewer for suggesting [5], a valuable and highly relevant reference. Their work demonstrates that poisoning large-scale training datasets is not merely a theoretical concern but a practical threat. They show how adversaries can inject poisoned examples into datasets at minimal cost by exploiting web-based data collection mechanisms.
>
> While we did not cite [5] in the current version of our paper, we have discussed related dataset poisoning studies in the background and related work. We will revise the paper to include [5] and clarify the differences between their work and ours.
>
> Unlike [5], which focuses on poisoning static datasets during pretraining, our attack operates in an iterative retraining setting, where models continuously adapt based on user feedback. Notably, our attacker does not require access to data collection pipelines or backend systems; instead, they can act entirely through public feedback mechanisms, such as voting or ranking systems.
>
> This makes our approach more practical and harder to detect, as it unfolds gradually over time without direct data manipulation. Whereas traditional poisoning attacks often aim to induce outright model failure, our objective is more subtle: gradually misaligning the model from genuine user preferences. In competitive settings, such gradual misalignment can be highly damaging while remaining difficult to trace.
>
> > "The conclusion feels a bit 'obvious'"
>
> While the conclusion that adversarial curation can lead to misalignment may seem intuitive, our work goes further by formally characterizing when and how this misalignment occurs. First, we clarify that such misalignment is not inevitable. According to Lemma 3.3, misalignment arises only under specific conditions: $\operatorname{Cov}\_{p\_{t}}\left[e^{r(x)},e^{\widetilde{r}\_{t}(x)}\right] < 0$. When $\operatorname{Cov}\_{p\_{t}}\left[e^{r(x)},e^{\widetilde{r}\_{t}(x)}\right] \geq 0$, adversial curation can still lead the model converging to the optimum that maximizes reward (at slower convergence rate); this is verified in Fig. 5 and highlights the inherent robustness of the iterative retraining process.
>
> Another less intuitive finding is that, while prior work on self-consuming generative models suggests that adding real data can effectively stabilize training [6, 7], we demonstrate that simply incorporating real data does not mitigate the effects of adversarial curated data.
>
> [1] Ouyang, L., Wu, J., Jiang, X., et al. Training language models to follow instructions with human feedback, 2022
>
> [2] Touvron, H., Martin, L., Stone, K., et al. Llama 2: Open foundation and fine-tuned chat models, 2023.
>
> [3] Kirstain, Y., Polyak, A., Singer, U., et al. Pick-a-pic: An open dataset of user preferences for text-to-image generation, 2023.
>
> [4] Liang, Y., He, J., Li, G., et al. Rich human feedback for text-to-image generation, 2024
>
> [5] Carlini, N., Jagielski, M., Choquette-Choo, C. A., et al. Poisoning web-scale training datasets is practical, 2024.
>
> [6] Alemohammad, S., Casco-Rodriguez, J., Luzi, L., et al. Self-consuming generative models go mad, 2023.
>
> [7] Bertrand, Q., Bose, A.J., Duplessis, A., et al. On the stability of iterative retraining of generative models on their own data, 2024.

---

### Decision · Program_Chairs · 2025-05-01

**Decision:**

Accept (poster)

**Comment:**

The authors present an extension of work on self-consuming generative models to the case of noisy or adversarially-curated data, and show that under certain conditions the models may not converge to the desired distribution. The contributions are well-motivated and tackle an important problem. The reviewers agreed that the work was high quality, and while some had concerns about the scale of experiments (limited to CIFAR-10 and synthetic distributions) and whether the results were surprising enough, all reviewers agreed that the work was worth accepting. I see no reason to disagree.